

# Global impact of mineral dust on cloud droplet number concentration

Vlassis A. Karydis[1], Alexandra P. Tsimpidi[1], Sara Bacer[1], Andrea Pozzer[1],
Athanasios Nenes[2,3,4] and Jos Lelieveld[1,5]

[1]Max Planck Institute for Chemistry, Mainz, 55128, DE
[2]Georgia Institute of Technology, Atlanta, GA, 30332, USA
[3]National Observatory of Athens, Palea Penteli, 15236, GR
[4]Foundation for Research and Technology Hellas, Patras, 26504, GR
[5]The Cyprus Institute, Nicosia, 1645, CY

## Abstract

The importance of wind-blown mineral dust for cloud droplet formation is studied by considering *i)* the adsorption of water on the surface of insoluble particles, *ii)* the particle coating by soluble material (due to atmospheric aging) which augments cloud condensation nuclei (CCN) activity, and *iii)* the effect of dust on inorganic aerosol concentrations through thermodynamic interactions with mineral cations. The ECHAM5/MESSy Atmospheric Chemistry (EMAC) model is used to simulate the composition of global atmospheric aerosol; the ISORROPIA-II thermodynamic equilibrium model treats the interactions of $K^+$-$Ca^{2+}$-$Mg^{2+}$-$NH_4^+$-$Na^+$-$SO_4^{2-}$-$NO_3^-$-$Cl^-$-$H_2O$ aerosol with gas-phase inorganic constituents. Dust is considered a mixture of inert material with reactive minerals; emissions are calculated online by taking into account the soil particle size distribution and chemical composition of different deserts worldwide. The impact of dust on droplet formation is treated through the "unified dust activation parameterization" that considers the inherent hydrophilicity from adsorption and acquired hygroscopicity from soluble salts during aging. Our simulations suggest that the presence of dust increases cloud droplet number concentrations (CDNC) over major deserts (e.g., up to 20% over the Sahara and Taklimakan Deserts) and decreases CDNC over polluted areas (e.g., up to 10% over southern Europe and 20% over northeastern Asia). This leads to a global net decrease of CDNC by 11%. The adsorption activation of insoluble aerosols and the mineral dust chemistry are shown to be equally important for the cloud droplet formation over the main desserts, e.g., by considering these effects CDNC increases by 20% over the Sahara. Remote from deserts the application of adsorption theory is critically important since the increased water uptake by the large aged dust particles (i.e., due to the added hydrophilicity by the soluble coating) reduce the maximum supersaturation



and thus the cloud droplet formation from the smaller anthropogenic particles (e.g.,
CDNC decreases by 10% over southern Europe and 20% over northeastern Asia by
applying adsorption theory). The global average CDNC decreases by 10% by
considering adsorption activation, while changes are negligible when accounting for
the mineral dust chemistry. Sensitivity simulations indicate that CDNC is also
sensitive to the mineral dust mass and inherent hydrophilicity, and not to the chemical
composition of the emitted dust.

**1. Introduction**
Atmospheric aerosols from anthropogenic and natural sources adversely affect
human health and influence the Earth's climate, both directly and indirectly
(Haywood and Boucher, 2000; Lohmann and Feichter, 2005; Andreae and Rosenfeld,
2008; IPCC, 2013; Kushta et al., 2014; Lelieveld et al., 2015). The direct climate
effect refers to the influence of aerosols on the radiative budget of Earth's atmosphere
by scattering and absorbing solar radiation (Seinfeld and Pandis, 2006). The indirect
effects include the ability of aerosols to affect the cloud optical thickness and
scattering properties of clouds (Twomey, 1974) as well as the cloud lifetime and
precipitation (Albrecht, 1989). The scientific interest in aerosol-cloud-climate
interactions initially focused on anthropogenic pollutants (e.g., sulfate) and to a lesser
extent on naturally emitted aerosols (e.g., sea salt). However, among atmospheric
aerosols, mineral dust is of particular importance since it is globally dominant in
terms of mass concentration in the atmosphere (Grini et al., 2005; Zender and Kwon,
2005) and can influence cloud and precipitation formation (Levin et al., 2005; Yin
and Chen, 2007; Karydis et al., 2011; Rosenfeld et al., 2011; Kallos et al., 2014).
Additionally, dust alone is responsible for more than 400,000 premature deaths per
year (Giannadaki et al., 2014).
Freshly emitted dust is considered insoluble. Reports of hygroscopic growth
measurements of dust particles indicate solubility to be very low, which together with
the observed cloud condensation nuclei (CCN), has been attributed to soluble ions
present in the particles (Gustafsson et al., 2005; Herich et al., 2009; Koehler et al.,
2009; Garimella et al., 2014). Chemistry – climate models (CCMs) typically use
Köhler theory to describe droplet formation from dust, which assumes that the CCN
activity depends solely on their curvature effect and the fraction of soluble material on
the particle (Smoydzin et al., 2012). However, mineral dust can adsorb water which





results in a surface film of water with reduced activity (Sorjamaa and Laaksonen,
2007), and promote the formation of cloud droplets at cloud-relevant supersaturation,
even of freshly emitted and chemically unprocessed dust particles (Sorjamaa and
Laaksonen, 2007; Kumar et al., 2009a).   Kumar et al. (2009a) emphasized the
importance of including water adsorption effects in describing the hygroscopic growth
of mineral aerosols, which was then included in a droplet formation parameterization
(Kumar et al. (2009b) for use in models. Evidence on the importance of adsorption
activation of dust particles is discussed in Kumar et al. (2011b; 2011a) for dry- and
wet-generated clays and mineral dusts representative of major regional dust sources
(North Africa, East Asia and North America). Adsorption activation was also found to
be important for volcanic ashes (Lathem et al., 2011). The observed hygroscopicity
could not be attributed to the soluble ions present, but rather to the strong water vapor
adsorption on the particle surface. Furthermore, the surface fractal dimension derived
from dust and ash critical supersaturation data agrees well with previous methods
based on measurements of nitrogen adsorption, which contribute strong evidence for
adsorption effects on water activity and droplet activation (Laaksonen et al., 2016),
despite concerns raised by Garimella et al. (2014) on multiple charging effects on the
work of Kumar et al. (2011b). Hatch et al. (2014) provided an alternative approach for
parameterizing CCN activation of fresh atmospheric mineral aerosol. This approach
was based on experimental water adsorption measurements on mineral clays
compared to CCN measurements used by Kumar et al. (2011b), which require
corrections for multiply charged particles and non-sphericity. Despite differences in
the adsorption parameters reported from the above two studies, the adsorption derived
CCN activities were quite similar and in excellent agreement. Based on these
findings, Karydis et al. (2011) integrated the Kumar et al. (2009b) parameterization
into the Global Modeling Initiative (GMI) chemical transport model (Considine et al.,
2005) and found that insoluble mineral dust can contribute up to 24% of the cloud
droplet number downwind of arid areas. Subsequently, the Kumar et al. (2009b)
parameterization has been integrated in a number of global and regional models and
applied to investigate the impact of mineral dust on warm cloud formation (Bangert et
al., 2012; Karydis et al., 2012; Gantt et al., 2014; Zhang et al., 2015).

104       Soluble inorganic ions like $Ca^{+2}$, $Mg^{+2}$, $Na^+$, and $K^+$ that exist on the surface of

mineral dust particles can participate in heterogeneous chemical reactions with acids
such as $HNO_3$ and $HCl$. Furthermore, dust particles can provide reaction sites for the



$SO_2$ oxidation into $H_2SO_4$. These processes result in the coating of dust particles by
soluble material, which augments the hygroscopicity of dust and therefore its ability
to act as CCN (Kelly et al., 2007). On the other hand, highly oxidized, soluble organic
species, particularly including carboxylic acid groups (e.g., oxalic acid), can interact
with particles dominated by di-valent salts (e.g., $CaCl_2$) and strongly decrease their
hygroscopicity (Drozd et al., 2014). Due to their relatively large size, chemically aged
dust particles can act as giant CCN, enhancing precipitation as they efficiently collect
moisture and grow at the expense of smaller droplets (Feingold et al., 1999; Levin et
al., 2005). In addition, giant CCN compete with the submicron particles for water
vapor, potentially reducing supersaturation and cloud droplet formation (Barahona et
al., 2010; Betancourt and Nenes, 2014b; Betancourt and Nenes, 2014a). Soluble
coatings on dust are mostly evident in the atmosphere after long-range transport of
dust plumes. Anthropogenic $NO_3^-$ and $SO_4^{2-}$ mainly contribute to the chemical aging
of dust over continents while sea salt derived $Cl^-$ is more important over oceans
(Sullivan et al., 2007; Fountoukis et al., 2009; Dall'Osto et al., 2010; Tobo et al.,
2010; Bougiatioti et al., 2016b; Weber et al., 2016). Apart from the gas phase
composition, the chemical processing of dust also depends on its chemical
composition and thus on the source region (Sullivan et al., 2009; Karydis et al., 2016).
Several studies have revealed that Saharan dust can be efficiently transported over the
Mediterranean basin where it can acquire significant soluble coatings (mostly sea salt
and sulfate) resulting in the enhancement of its hygroscopicity and CCN activity
(Wurzler et al., 2000; Falkovich et al., 2001; Smoydzin et al., 2012; Abdelkader et al.,
2015). Twohy et al. (2009) have shown that Saharan dust often acts as CCN over the
eastern North Atlantic and significantly contributes to cloud formation west of Africa.
Begue et al. (2015) analyzed a case of possible mixing of European pollution aerosols
with Saharan dust transported over northern Europe, and found that aged Saharan dust
was sufficiently soluble to impact the hygroscopic growth and cloud droplet
activation over the Netherlands. Asian dust has also been reported to have a
considerable impact on cloud formation after being transported over long distances
and mixed with soluble materials (Perry et al., 2004; Roberts et al., 2006; Sullivan et
al., 2007; Ma et al., 2010; Stone et al., 2011; Yamashita et al., 2011).
Despite the importance of mineral dust aerosol chemistry for accurately predicting
the aerosol hygroscopicity changes that accompany these reactions, most
thermodynamic models used in global studies lack a realistic treatment of crustal



species, e.g., assuming that mineral dust is chemically inert (Liao et al., 2003; Martin et al., 2003; Koch et al., 2011; Leibensperger et al., 2011). Only few global studies have accounted for the thermodynamic interactions of crustal elements with inorganic aerosol components (Feng and Penner, 2007; Fairlie et al., 2010; Xu and Penner, 2012; Hauglustaine et al., 2014; Karydis et al., 2016). Most of these models either neglect the impact of dust on cloud droplet formation or apply simplified assumptions about the CCN activity of dust, e.g., they convert "hydrophobic" dust to "hydrophilic" dust by applying a constant κ-hygroscopicity (e.g., 0.1) and use Köhler theory to describe cloud droplet activation. However, accounting for both the inherent hydrophilicity of dust and the acquired hygroscopicity from soluble salts could improve the predictive capability of CCMs. For this purpose, Kumar et al. (2011a) presented a "unified dust activation framework" (UAF) to treat the activation of dust with substantial amounts of soluble material by considering the effects of adsorption (due to the hydrophilicity of the insoluble core) and absorption (due to the hygroscopicity of the soluble coating) on CCN activity. Karydis et al. (2011) provided a first estimate of aged dust contribution to global CCN and cloud droplet number concentration (CDNC) by using the UAF. They found that coating of dust by hygroscopic salts can cause a twofold enhancement of its contribution to CCN. On the other hand, aged dust can be substantially depleted due to in-cloud supersaturation and eventually reduce the CDNC. Bangert et al (2012) investigated the impact of Saharan dust on cloud droplet formation over western Europe and found only a slight increase in calculated CDNC. However, these studies did not include thermodynamic interactions of mineral dust with sea salt and anthropogenic pollutants. Instead, a prescribed fraction of mineral dust that is coated with ammonium sulfate was used to represent the aged dust.

The present work aims at advancing previous studies on dust influences of cloud droplet formation by comprehensively considering *i*) the adsorption of water on the surface of insoluble dust particles, *ii*) the coating of soluble material on the surface of mineral particles which augments their CCN activity, and, *iii*) the effects of dust on the inorganic soluble fraction of dust through thermodynamic interactions of semi-volatile inorganic species and sulfate with mineral cations. The ECHAM5/MESSy Atmospheric Chemistry (EMAC) model (Jöckel et al., 2006) is used to simulate aerosol processes, while the "unified dust activation framework" (Karydis et al., 2011; Kumar et al., 2011a) is applied to calculate the CCN spectra and droplet number



concentration, by explicitly accounting for the inherent hydrophilicity from adsorption
and acquired hygroscopicity from soluble salts by dust particles from atmospheric
aging. Mineral dust chemistry has been taken into account by using the
thermodynamic equilibrium model ISORROPIA II (Fountoukis and Nenes, 2007).
Dust emissions are calculated online by an advanced dust emission scheme which
accounts for the soil particle size distribution (Astitha et al., 2012) and chemical
composition (Karydis et al., 2016) of different deserts worldwide. The sensitivity of
the simulations to the emitted dust aerosol load, the mineral dust chemical
composition and the inherent hydrophilicity of mineral dust is also considered.
**2.  Model Description**
**2.1 EMAC Model**
We used the ECHAM5/MESSy Atmospheric Chemistry (EMAC) model (Jöckel et
al., 2006) which uses the Modular Earth Submodel System (MESSy2) (Jöckel et al.,
2010) to connect submodels that describe the lower and middle atmosphere processes
with the 5th generation European Centre - Hamburg (ECHAM5) general circulation
model (GCM) as a dynamical core (Röckner et al., 2006). EMAC has been
extensively described and evaluated against in-situ observations and satellite
retrievals (de Meij et al., 2012; Pozzer et al., 2012; Tsimpidi et al., 2014; Karydis et
al., 2016). The spectral resolution of the EMAC model used in this study is T63L31,
corresponding to a horizontal grid resolution of approximately 1.9º×1.9º and 31
vertical layers between the surface and 10 hPa (i.e. 25 km altitude). EMAC is applied
for 2 years covering the period 2004-2005 and the first year is used as spin-up.
EMAC simulates the gas phase species through the MECCA submodel (Sander et
al., 2011). Aerosol microphysics are calculated by the GMXe module (Pringle et al.,
2010). The organic aerosol formation and chemical aging is calculated by the
ORACLE submodel (Tsimpidi et al., 2014). The CLOUD submodel (Röckner et al.,
2006) calculates the cloud cover as well as cloud micro-physics and precipitation of
large scale clouds (i.e., excluding convective clouds). The wet and dry deposition are
calculated by the SCAV (Tost et al., 2006) and the DRYDEP (Kerkweg et al., 2006)
sub-models.
The inorganic aerosol composition is computed with the ISORROPIA-II
(http://isorropia.eas.gatech.edu) thermodynamic equilibrium model (Fountoukis and





Nenes, 2007) with updates as discussed in Capps et al. (2012). ISORROPIA-II
calculates the gas-liquid-solid equilibrium partitioning of the $K^+$-$Ca^{2+}$-$Mg^{2+}$-$NH_4^+$-
$Na^+$-$SO_4^{2-}$-$NO_3^-$-$Cl^-$-$H_2O$ aerosol system. Potassium, calcium, magnesium, and
sodium are assumed to exist in the form of $Ca(NO_3)_2$, $CaCl_2$, $CaSO_4$, $KHSO_4$, $K_2SO_4$,
$KNO_3$, $KCl$, $MgSO_4$, $Mg(NO_3)_2$, $MgCl_2$, $NaHSO_4$, $Na_2SO_4$, $NaNO_3$, $NaCl$ in the solid
phase and $Ca^{2+}$, $K^+$, $Mg^{2+}$, $Na^+$ in the aqueous phase. More details about the EMAC
model set up used in this study can be found in Karydis et al. (2016).

**2.2 CCN Activity and Cloud Droplet Formation Parameterization**
The equilibrium supersaturation, $s$, over the surface of a water droplet containing a
solute particle (i.e., without any insoluble material present) is calculated using the
hygroscopicity parameter, $\kappa$, based on $\kappa$-Köhler theory (Petters and Kreidenweis,

221   2007):

$$s = \frac{4\sigma M_w}{RT\rho_w D_P} - \frac{D_{dry}^3 \kappa}{D_P^3} \quad (1)$$


where $D_{dry}$ is the dry CCN diameter, $D_p$ is the droplet diameter, $\sigma$ is the CCN surface
tension at the point of activation, $\rho_w$ is the water density, $M_w$ is the molar mass of
water, $R$ is the universal gas constant, and $T$ is the average column temperature.
For insoluble particles (e.g., pristine mineral dust), the multilayer Frenkel-Halsey-
Hill (FHH) adsorption isotherm model (Sorjamaa and Laaksonen, 2007) is used,
which contains two adjustable parameters ($A_{FHH}$ and $B_{FHH}$) that describe the
contribution of water vapor adsorption on CCN activity. In this case, the equation
describing the equilibrium supersaturation over the surface of a water droplet is given
by (Kumar et al., 2009b):

$$s = \frac{4\sigma M_w}{RT\rho_w D_P} - A_{FHH}\left(\frac{D_P - D_{dry}}{2D_w}\right)^{-B_{FHH}}$$

232   (2)

where $D_w$ is the diameter of a water molecule. The adsorption parameter $A_{FHH}$
represents the interactions between the first water monolayer and the dust surface.
$B_{FHH}$ expresses the long range interactions of additional adsorbed water layers with
the dust surface. Kumar et al. (2011b) tested a wide range of fresh unprocessed
regional dust samples and minerals and found that one set of the FHH parameters



($A_{FHH}$ =2.25±0.75, $B_{FHH}$ =1.20±0.10) adequately reproduces the measured CCN
activity for all dust types considered.
To account for the coating of soluble material on the surface of mineral dust, the
"unified activation framework" (Karydis et al., 2011; Kumar et al., 2011a) is used,
which describes the water vapor supersaturation over an aerosol particle consisting of
insoluble core with a soluble coating:
$$s = \frac{4\sigma M_w}{RT\rho_w D_P} - \frac{\varepsilon_s D_{dry}^3 \kappa}{\left(D_P^3 - \varepsilon_i D_{dry}^3\right)} - A_{FHH}\left(\frac{D_P - \varepsilon_i^{1/3} D_{dry}}{2D_w}\right)^{-B_{FHH}} \quad (3)$$
where $\varepsilon_i$ is the insoluble volume fraction and $\varepsilon_s$ is the soluble volume fraction. Eq. 3
takes into account both the inherent hydrophilicity from adsorption expressed in the
third term of the equation and the acquired hygroscopicity from soluble salts by dust
particles expressed in the second term of the equation. The first term accounts for the
Kelvin effect. Noting that for a complete insoluble dust particle, i.e., as $\varepsilon_s \rightarrow 0$ and
$\varepsilon_i \rightarrow 1$, the UAF approaches FHH theory (Eq. 2).
Calculation of CDNC is carried out in two conceptual steps, one involving the
determination of the "CCN spectrum" (i.e., the number of CCN that can activate to
form droplets at a certain level of supersaturation), and another one determining the
maximum supersaturation, $s_{max}$, that develops in the ascending cloudy air parcels used
to represent droplet formation in EMAC. The CDNC is then the value of the CCN
spectrum at $s_{max}$.
The "CCN spectrum", $F^s(s)$, is computed following Kumar et al. (2009b) and
assumes that particles can be described either by KT or FHH theory. $F^s(s)$ for an
external mixture of lognormal particle size distributions is given by:
$$F^s(s) = \int_0^s n^s(s)ds = \sum_{i=1}^{n_m} \frac{N_i}{2} erfc\left[-\frac{\ln\left(\frac{s_{g,i}}{s}\right)}{x\sqrt{2}\ln(\sigma_i)}\right] \quad (4)$$
where $s$ is the level of water vapor supersaturation, $n^s(s)$ is the critical
supersaturation distribution, $s_{g,i}$ is the critical supersaturation of the particle with a
diameter equal to the geometric mean diameter of the mode $i$, $\sigma_i$ is the geometric





standard deviation for the mode $i$, and $x$ is an exponent that depends on the
activation theory used. For modes following Köhler theory, $x = -\dfrac{3}{2}$ (Fountoukis and
Nenes, 2005), while for insoluble particles following FHH theory, $x$ depends on
$A_{FHH}$ and $B_{FHH}$ (Kumar et al., 2009b). In the case of UAF $x$ lies between the KT
and FHH-AT limits, and is determined from Eq. (3) by performing a power law fit
between $s_g$ and $D_{dry}$ as described in Kumar et al. (2011a). The calculation of $s_g$
involves determining the maximum of the relevant equilibrium curve in equilibrium
with the surrounding water vapor ($\left. \dfrac{ds}{dD_p} \right|_{D_p = D_g} = 0$ in Eqs. 1-3). Once $D_g$ is determined,
it can be substituted in Eqs. 1-3 to obtain $s_g$.

273        The maximum supersaturation, $s_{\max}$, in the ascending parcel is calculated from an

equation that expresses the supersaturation tendency in cloudy air parcels, which at
the point of maximum supersaturation becomes (Nenes and Seinfeld, 2003; Barahona
and Nenes, 2007)
$$\frac{2aV}{\pi \gamma \rho_w} - G s_{\max} I\left(0, s_{\max}\right) = 0 \quad (5)$$

where $V$ is the updraft velocity (i.e., not including convection) calculated online by
assuming that the sub-grid vertical velocity variability is dominated by the turbulent
transports and by choosing the root-mean-square value of the GCM model-generated
turbulent kinetic energy ($TKE$) as a measure. Based on this assumption, the in-cloud
updraft velocity can be expressed as $V = {}^-V + 0.7\ \sqrt{TKE}$, where ${}^-V$ is the GCM-
resolved large scale updraft velocity (Lohmann et al., 1999a; Lohmann et al., 1999b).
Following Morales and Nenes (2010), $V$ can be considered as a "characteristic updraft
velocity" which yields CDNC value representative of integration over a probability
density function (PDF) of updraft velocity. Morales and Nenes (2010) have shown
that this assumption applies well to large scale clouds (i.e., stratocumulus), which are
the type of clouds described by the CLOUD sub-model in EMAC. $a, \gamma, G$ in Eq. (5)
are parameters defined in Nenes and Seinfeld (2003). $I(0, s_{\max})$ is the "condensation
integral" which expresses the condensational depletion of supersaturation upon the
growing droplets at the point of $s_{\max}$ in the cloud updraft. It is expressed as the sum of
two terms:






$$I\left(0, s_{\max}\right) = I_K\left(0, s_{\max}\right) + I_{FHH}\left(0, s_{\max}\right) \quad (6)$$



The first term on the right hand side of Eq. (6), $I_K\left(0, s_{\max}\right)$, describes the contribution
from particles that follow the Köhler theory and is calculated using the revisited
population splitting approach of Betancourt and Nenes (2014a). The second term,
$I_{FHH}\left(0, s_{\max}\right)$, represents the contribution of freshly emitted or aged dust particles to
the condensation integral and is represented in Kumar et al. (2009b) and Karydis et al.
(2011). Once $s_{\max}$ is determined by numerically solving Eq. (5), the number of cloud
droplets that form in the parcel, $N_d$, is obtained from the "CCN spectrum" (Eq. (4))
computed for $s_{\max}$, i.e., $N_d = F\left(s_{\max}\right)$.

**2.3 Aerosol Precursor Emissions**
Dust emission fluxes are calculated online by an advanced dust flux scheme
developed by Astitha et al. (2012). This scheme uses an explicit geographical
representation of the airborne soil particle size distribution based on soil
characteristics in every grid cell. Emissions of crustal species ($Ca^{2+}$, $Mg^{2+}$, $K^+$, $Na^+$)
are estimated as a fraction of mineral dust emissions based on the chemical
composition of the emitted soil particles in every grid cell (Karydis et al., 2016).
Emissions of sea spray aerosols are based on the offline monthly emission data set of
AEROCOM (Dentener et al., 2006) assuming a composition of 55% $Cl^-$, 30.6% $Na^+$,
7.7% $SO_4^{2-}$, 3.7% $Mg^{2+}$, 1.2% $Ca^{2+}$, 1.1% $K^+$ (Seinfeld and Pandis, 2006). The
CMIP5 RCP4.5 emission inventory (Clarke et al., 2007) is used for the anthropogenic
primary organic aerosol emissions from fossil fuel and biofuel combustion sources.
The open biomass burning emissions from savanna and forest fires are based on the
GFED v3.1 database (van der Werf et al., 2010). More details about the aerosol phase
emissions used by EMAC can be found in Karydis et al. (2016) and Tsimpidi et al.

320 (2016).

Related anthropogenic emissions of $NO_x$, $NH_3$, and $SO_2$, which represent the
gaseous precursors of the major inorganic components, are based on the monthly
emission inventory of EDGAR-CIRCE (Doering, 2009) distributed vertically as
presented in Pozzer et al. (2009). The natural emissions of $NH_3$ are based on the





GEIA database (Bouwman et al., 1997). $NO_x$ produced by lightning is calculated
online and distributed vertically based on the parameterization of Grewe et al. (2001).
The emissions of NO from soils are calculated online based on the algorithm of
Yienger and Levy (1995) as described in Ganzeveld et al. (2002). Eruptive and non-
eruptive volcanic degassing emissions of $SO_2$ are based on the AEROCOM data set
(Dentener et al., 2006). The oceanic DMS emissions are calculated online by the
AIRSEA submodel (Pozzer et al., 2006). More details about the gas phase emissions
used by EMAC can be found in Pozzer et al. (2012) and Karydis et al. (2016).

**3. Model Results and Evaluation**
**3.1 Model Predictions**
The annual and seasonal (during DJF and JJA) mean CDNC calculated by EMAC
with UAF implementation for the lowest model level at which clouds are formed
(centered at 940 mb) are shown in Figure 1. The calculated CDNC is mostly sensitive
to the cloud updraft velocity and the total aerosol number concentration (Karydis et
al., 2012), which are the main drivers of the $s_{max}$ calculations. The annual mean
aerosol number concentration, updraft velocity, and $s_{max}$ calculated by EMAC at 940
mb are shown in Figure 2. The calculated global annual mean CDNC at 940 mb is
231 $cm^{-3}$.
Over the continents, the predicted annual mean CDNC is 546 $cm^{-3}$ and exceeds
1000 $cm^{-3}$ over the industrialized areas of Europe, central and eastern Asia, and North
America. In these areas, the aerosol number concentration is high (exceeding 10,000
$cm^{-3}$; Figure 2a), while the calculated updraft velocities (0.5-1 m $s^{-1}$; Figure 2b) allow
the development of sufficiently high $s_{max}$ (0.1-0.3%; Figure 2c) for the activation of
5% (over eastern China) to 15% (over central Europe) of the pollution aerosols into
cloud droplets. The simulated $s_{max}$ is close to the estimated $s_{max}$ (0.2%-0.5%) for
stratocumulus clouds based on data from continental air masses (Twomey and
Wojciechowski, 1968; Martin et al., 1993) indicating that the combination of aerosol
number concentration and updraft velocity in the model is realistic. While the aerosol
number concentration over the industrialized areas remains fairly constant throughout
the year, the updraft velocity is higher during the boreal winter (i.e., DJF) resulting in
a seasonal peak of CDNC during DJF (exceeding 2,000 $cm^{-3}$) over North America,





Europe and eastern Asia (Figure 1b). The highest annual mean CDNC is calculated
over northern India (~2,000 cm$^{-3}$) where the model simulates highest aerosol
concentrations (~30,000 cm$^{-3}$). Over Southeast Asia and India, CDNC peaks during
JJA (exceeding 2,000 cm$^{-3}$; Figure 1c), affected by the East Asian Monsoon and the
high updraft velocities developed during the wet season. Relatively high CDNC
(annual mean of 300-700 cm$^{-3}$) are also calculated over the tropical regions of the
Southern Hemisphere which are influenced by biomass burning. Relatively low
values are calculated over the Congo Basin where the mean updraft velocity is
typically low (below 0.2 m s$^{-1}$) leading to low $s_{max}$ (below 0.05%) and cloud droplet
activation (~300 cm$^{-3}$). Downwind of deserts, the calculated CDNC varies between
100 cm$^{-3}$ (e.g., Patagonia, and Australian deserts) to 1,000 cm$^{-3}$ (e.g., Sahara, Arabian,
Taklimakan, Gobi and Atacama). In the vicinity of the Sahara and Arabian deserts,
the mean updraft velocity is ~0.5 m s$^{-1}$. However, downwind of the western part of the
Sahara the aerosol number concentration is relatively low (~1,000 cm$^{-3}$) leading to
higher $s_{max}$ (~0.2%) but low CDNC (~200 cm$^{-3}$). On the other hand, downwind of the
eastern Sahara and Arabian deserts the aerosol concentration is higher (2,000-3,000
cm$^{-3}$). Over these areas the presence of a high number of coarse dust particles
significantly reduces $s_{max}$ (~0.05%), but at the same time they efficiently activate into
cloud droplets (CDNC varies from 500 to 1,000 cm$^{-3}$). Close to Patagonia and
Australia, despite the high updraft velocities (~1 m s$^{-1}$), the aerosol concentration is
low (below 500 cm$^{-3}$) and also CDNC is relatively low (~100 cm$^{-3}$). The highest
updraft velocities are calculated around the Atacama and Gobi deserts (over 1 m s$^{-1}$)
leading to both high $s_{max}$ (over 0.3%) and CDNC (~1,000 cm$^{-3}$). However, the central
Asian deserts (e.g., Gobi) are under the influence of the Siberian anticyclone during
winter (i.e., DJF) which causes katabatic winds (that inhibit the formation of positive
updraft velocities) and very low temperatures that prevent the formation of liquid
clouds.
Over the oceans, the predicted annual mean CDNC is 113 cm$^{-3}$ and exceeds 500
cm$^{-3}$ along the coasts of Mediterranean countries, China, India, SE Asia, California,
the northeastern USA and western Africa (Fig. 1). Over many coastal regions aerosol
concentrations are relatively high (5,000-10,000 cm$^{-3}$), however, the low updraft
velocities (~0.2 m s$^{-1}$) result in lower CDNCs than over land (Figure 1). The
Mediterranean and Yellow Seas are somewhat exceptional since the annual mean



updraft velocities are higher (~0.3 m s$^{-1}$), resulting in higher $s_{max}$ (~0.1% and ~0.3%,
respectively) and therefore high CDNC (~800 cm$^{-3}$ and ~1200 cm$^{-3}$, respectively).
The simulated $s_{max}$ is in close agreement with estimates (~0.1%) based on
observational data over the eastern Mediterranean (Bougiatioti et al., 2016a;
Kalkavouras et al., 2016). CDNC over these seas is subject to high seasonal variation
ranging from ~400 cm$^{-3}$ (~800 cm$^{-3}$) over the Mediterranean (Yellow) Sea during JJA,
to over 1,000 cm$^{-3}$ (2,000 cm$^{-3}$) during DJF due to the higher updraft velocities during
boreal winter (exceeding 1 m s$^{-1}$) compared to summer (below 0.2 m s$^{-1}$). Over the
northern coasts, the annual mean CDNC is significantly enhanced compared to the
oceans of the Southern Hemisphere due to the transport of pollutants from
industrialized areas in the Northern Hemisphere. Despite the high updraft velocities
calculated over the southern oceans throughout the year (up to 1 m s$^{-1}$), the lack of
aerosol (typically below 100 cm$^{-3}$) results in CDNC below 50 cm$^{-3}$. Finally, the
calculated CDNC decreases with altitude due to the decrease in aerosol concentration
by dilution and atmospheric removal (Figure 3). The global mean CDNC is predicted
to be 231 cm$^{-3}$, 171 cm$^{-3}$, 120 cm$^{-3}$, 87 cm$^{-3}$, and 60 cm$^{-3}$ at 940 mb, 900 mb, 860 mb,
820 mb, and 770 mb, respectively.

**3.2    Model Evaluation**
The predicted CDNC are compared to observational data from continental,
polluted marine and clean marine regions around the world (Karydis et al., 2011). The
locations of observations (i.e., longitude, latitude, and altitude) and time of year have
been taken into account in sampling the model results. Given that the observations
span a decade, in contrast to the simulation which represents one year, the month of
each campaign has been used to account for the seasonal variability of the CDNC.
Thus, the implicit assumption is that inter-annual variability can be neglected. It
should also be mentioned that the observations typically do not represent monthly
means over 1.9° grid squares, as sampled from the model results, so that the
comparison is more qualitative than quantitative. A summary of the comparison
results is presented in Table 1 and Figure 4. The mean bias (MB), mean absolute
gross error (MAGE), normalized mean bias (NMB), normalized mean error (NME),
and the root mean square error (RMSE) are used to assess the model performance
(Table 2).



The model captures the low values (below 100 cm$^{-3}$) observed over the remote
Pacific, Atlantic and Indian Oceans and at the same time is capable of simulating the
higher concentrations (>100 cm$^{-3}$) observed over the eastern Pacific Ocean (Table 1).
On the other hand, it falls short in reproducing the relatively high CDNC (>100 cm$^{-3}$)
observed during summer over the western Arctic Ocean and over the remote area west
of Australia. Overall, the model tends to underestimate the CDNC over remote oceans
with a MB = -33 cm$^{-3}$ and NMB = -39% (Table 2).
Both the observed and simulated CDNC show significant increases over polluted
marine regions close to the coasts (Table 1; Figure 4a). Compared to satellite
retrievals (Bennartz, 2007; Rausch et al., 2010), the model reproduces the CDNC over
the American and African coasts well, but it significantly overestimates CDNC along
the Asian coasts (Table 1). Compared to in situ observations, the model reproduces
the high CDNC along coastal areas in the Northern Hemisphere (e.g., the Yellow Sea,
Oregon, Florida, Canary Islands), but systematically overestimates CDNC over the
British coasts. Further, the model does not reproduce some of the high CDNC
observations over more remote areas (i.e, over the Azores and eastern Atlantic
Ocean). Overall, the model tends to overestimate the CDNC over polluted marine
areas with a MB = 127 cm$^{-3}$ and NMB = 75% (Table 2).
The observed CDNC over continental regions is subject to high variability, with
reported values ranging from <100 cm$^{-3}$ over Alaska (Dong and Mace, 2003) to
>1,000 cm$^{-3}$ over China (Zhao et al., 2006), England (Bower et al., 1999), and the
continental USA (Fountoukis et al., 2007). The model captures the observed
variability with low values over remote areas (e.g., over Alaska) and high values over
the industrialized parts of the Northern Hemisphere (i.e., East Asia, Europe, and
China). Overall, the model overestimates CDNC over most regions (MB= 269 cm$^{-3}$
and NMB=58%; Table 2). Over China, the simulated CDNC is within the observed
range with the exception of Hebei Province where it significantly overestimates
measured CDNC (Table 1). In Europe, the model reproduces the high CDNC
observed over Central Europe and England but it clearly overestimates the low CDNC
values observed over Finland. Over North America, the model captures the variability
of the observed CDNC, predicting lower values over remote areas (e.g., Alaska) and
higher values over the industrialized areas of USA (e.g., Ohio and Michigan). It tends
to overestimate the CDNC over the continental USA and underestimate the observed
values over Alaska.



Over all examined regions (clean marine, polluted marine, continental), the
calculated NMB is 56% and the NME is 82%, indicating that some of the discrepancy
between the modelled and the observed CDNC is explained by uncertainties in the
observations and the numerical simulations. Around 60% of the simulated CDNC are
within a factor of 2 compared to the measurements (Figure 4a) and 40% of the
simulated CDNC differ less than 30% from the measurements. Based on the typical
properties of marine stratus clouds, a uniform increase in global CDNC by 30% (or
50%) can result in a perturbation of -1.1 W m$^{-2}$ (or -1.7 W m$^{-2}$) in the global mean
cloud radiative forcing (Schwartz, 1996). However, the simulated CDNC presented
here refers to the number concentration of droplets nucleated in clouds and represents
an upper limit with respect to the comparison with observations, since collision and
coalescence processes, which are not taken into account here, can reduce the CDNC.

**4. Mineral Dust Effect on CDNC**

**4.1 Total Impact of Mineral Dust on CDNC**
To estimate the overall effect of mineral dust on CDNC a sensitivity run was
conducted switching off the mineral dust emissions. Figure 5 depicts the difference in
CDNC between the base case simulation and the sensitivity test. A positive change
corresponds to an increase of the CDNC due to the presence of dust. The predicted
CDNC is typically increased by the presence of dust aerosols over the main deserts
(Figure 5). Over the Sahara, CDNC increases less than 50 cm$^{-3}$ (up to 20%). The
largest change is calculated downwind of the Patagonian (~150 cm$^{-3}$ or 70%) and
Atacama (~350 cm$^{-3}$ or 40%) deserts. Over these deserts dust emissions increase the
aerosol concentration by more than 5,000 cm$^{-3}$. The effect of mineral dust on CDNC
close to Sahara varies significantly throughout the year due to the seasonality of the
mineral dust emissions. Over the sub-Sahelian region, CDNC increases by up to 150
cm$^{-3}$ during DJF, owing to the northeasterly trade winds (i.e., Harmattan winds) which
blow from the Sahara Desert over the West Africa during winter. Over the eastern
Sahara and the Arabian deserts CDNC increase up to 150 cm$^{-3}$ during spring (i.e.,
MAM) and autumn (i.e, SON) when the Sirocco winds are most common. In contrast
to regions close to deserts, CDNC decreases over the polluted regions of the Northern
Hemisphere and especially over southern Europe (~100 or less than 10%) and
northeastern Asia (up to 400 cm$^{-3}$ or 20%). In these areas, dust particles transported



from the Sahara over Europe and from the Gobi and Taklimakan deserts over Asia,
are mixed with anthropogenic particles affecting the aerosol-water vapor interactions.
As the insoluble fraction of aerosols increases, the exponent $x$ in Eq. 4 changes,
resulting in a decrease of the number of activated droplets. Furthermore, the relatively
large, aged dust particles over these areas activate early on in the cloud formation
process, taking up much water per particle and thus reducing $s_{max}$ (~15%), and
consequently cloud droplet formation on the smaller anthropogenic particles (e.g., the
activated fraction of the particles in the accumulation mode reduces by 20%). Beside
microphysical effects, the presence of mineral dust can also affect cloud formation by
altering the energy balance of the atmosphere, and thus turbulent motions and the
updraft velocity. Nevertheless, the calculated updraft velocity does not change
significantly between the two simulations (less than 5%) since the meteorology is
dynamically nudged to analysis data (Jeuken et al., 1996). CDNC also decreases over
the oceans downwind of deserts in the Northern Hemisphere, and even over the
rainforests in the Southern Hemisphere (~150 or 30%). Overall, despite that CDNC
increases over the deserts due to the presence of dust particles, the decrease of CDNC
over the industrialized and forested continental areas dominates the calculated global
average change, i.e., the calculated global average CDNC decreases by 11% (or 26
$cm^{-3}$).

## 4.2 Impact of Mineral Dust Chemistry on CDNC

To estimate the effects of thermodynamic mineral dust interactions with inorganic
anions on the predicted CDNC, a sensitivity run was conducted by switching off the
dust-aerosol chemistry. Karydis et al. (2016) have shown that dust can significantly
affect the partitioning of inorganic aerosol components and especially nitrate.
Analogous to (Karydis et al. (2016)), accounting for thermodynamic interactions of
mineral dust in our simulations results in an increase of the tropospheric burden of
nitrate, chloride, and sulfate aerosols by 44%, 9%, and 7%, respectively. On the other
hand, ammonium decreases by 41%. The dust presence itself also decreases by 14%
since it becomes significantly more soluble, mostly due to the condensation of nitric
acid on its surface, and is removed more efficiently through wet and dry deposition,
the latter due to the increased sedimentation by dust particles that have a larger water
content. Therefore, the calculated change of CDNC (Figures 6a and 6b) is the net


result of counterbalancing effects. Due to the increase of the soluble fraction by
considering mineral dust chemistry, the CDNC activated from dust particles increases
(Figure 6c), while the total number of dust particles and the CDNC from insoluble
particles decreases (Figure 6d). Taking as an example a grid cell over the Sahara
desert, the model simulations indicate that by switching on the mineral dust
chemistry, the soluble fraction of the dust containing particles increases by 0.07,
resulting in an increase of CDNC activated from soluble aerosol modes by 150 cm$^{-3}$
(Figure 6c). On the other hand, the aerosol number concentration decreases by 90 cm$^{-3}$
due to the more efficient atmospheric removal of the aged dust particles, resulting in
a decrease of the CDNC activated from the insoluble modes by 50 cm$^{-3}$ (Figure 6d).
The net effect is that the total CDNC increases by 100 cm$^{-3}$ (Figure 6a).
Overall, the presence of reactive dust components results in an increase of CDNC
over the deserts that are close to anthropogenic sources, e.g., up to 100 cm$^{-3}$ (or 20%)
over the Sahara and up to 200 cm$^{-3}$ (or 30%) over the Arabian Peninsula. In these
areas, the CCN activity of mineral dust (initially hydrophilic) is enhanced by the
acquired hygroscopicity from the anthropogenic (including biomass burning) aerosol
compounds (mainly nitrate) during their thermodynamic interaction. Even though the
chemically aged dust particles activate into droplets more efficiently than insoluble
ones, their reduced number concentration dominates the calculated effect on CDNC
over the relatively pristine remote desert regions, e.g., CDNC decreases up to 200 cm$^{-3}$
(or 20%) downwind of the Taklimakan, 250 cm$^{-3}$ (or 30%) around the Atacama, and
up to 100 cm$^{-3}$ (or 40%) over the Patagonian deserts. Even over the rainforests, HNO$_3$
from biomass burning NO$_X$ thermodynamically interacts with the coarse soil particles
from the upwind deserts, resulting in an increase of CDNC by around 50 cm$^{-3}$. CDNC
is also slightly increased over Europe and eastern Asia (up to 150 cm$^{-3}$ or about 10%)
where HNO$_3$ from anthropogenic NO$_X$ sources interacts with mineral dust from the
surrounding deserts. While the global average CDNC does not change much by taking
into account thermodynamic and chemical interactions of mineral dust with inorganic
air pollutants, CDNC spatial distributions change substantially.

**4.3 Impact of Water Adsorption by Mineral Dust on CDNC**
To estimate the effects of water adsorption onto the surface of insoluble dust
particles on CDNC, a sensitivity run was conducted by switching off the FHH
adsorption calculations. In this sensitivity simulation, the soluble modes follow the κ-



Köhler theory while insoluble modes do not participate in cloud droplet formation
calculations. Figure 7 depicts the difference in CDNC between the base case
simulation and this sensitivity test. A positive change corresponds to an increase of
the CDNC from water adsorption on mineral dust. The calculations show that CDNC
is increased by applying FHH theory over several arid areas where the insoluble dust
concentration is high (Figure 7), since κ-Köhler theory does not take into account the
contribution of insoluble particles to cloud droplet formation. CDNC is increased in
the vicinity of the Sahara, Arabian and Thar deserts (~100 cm$^{-3}$ or about 20%) where
the insoluble fraction of mineral dust is larger due to the small anthropogenic
emission influence that makes the particles hygroscopic. On the other hand, CDNC
decreases over the polluted regions of the Northern Hemisphere and especially over
Europe (~100 cm$^{-3}$ or about 10%) and Asia (up to 400 cm$^{-3}$ or ~20%). Over these
areas, the added hydrophilicity by the soluble coatings on the surface of the aged dust
particles increases their water uptake during activation. Therefore, the aged dust
particles relatively strongly compete for water vapor, reducing the $s_{max}$ (~15%) and
thus cloud droplet formation from the smaller anthropogenic particles. Over the
tropical rainforests CDNC decreases by approximately 150 cm$^{-3}$ (or ~30%). Overall,
the use of the UAF results in a decrease of the global average CDNC by ~10% (or
about 23 cm$^{-3}$).

## 5    Additional Sensitivity Tests

Three additional sensitivity simulations were conducted to investigate the CDNC
dependency on i) the chemical composition of the emitted dust aerosols, ii) the
hydrophilicity of mineral dust, and iii) the strength of the dust aerosol emissions.
Figure 8 depicts the absolute annual mean changes in CDNC compared to the
reference simulation for each of the sensitivity tests. A positive change corresponds to
an increase of the CDNC relative to the reference.

### 5.1 Sensitivity to the emitted dust aerosol composition

The first sensitivity test assumes a globally uniform chemical composition of
mineral dust (Sposito, 1989), in contrast to the reference simulation where the mineral
dust composition depends on the soil characteristics of each desert (Karydis et al.,
2016). While the emitted mineral dust load remains the same in the sensitivity



simulation, the different mineral dust composition results in significant changes in the
calculated tropospheric burdens of dust components (Karydis et al., 2016). In
particular, the fraction of the mineral components relative to the total dust in the
sensitivity simulation is lower over most of the deserts compared to the reference.
This reduction of the chemically reactive mineral components in the sensitivity
simulation results in a slowdown of the mineral dust aging and hence in an increase of
its concentration due to the reduced atmospheric removal. Conversely, the CCN
activity of dust particles is higher in the reference simulation since the chemical aging
is stronger compared to the sensitivity simulation. These counterbalancing effects
result in negligible changes of CDNC worldwide (less than 10%).



**5.2 Sensitivity to the hydrophilicity of dust**
The second sensitivity test assumes increased hydrophilicity of mineral dust
aerosols by using a 10% lower $B_{FHH}$ parameter ($B_{FHH}$=1.1). The higher hydrophilicity
of mineral dust in the sensitivity simulation results in increased CDNC over over
areas close to deserts by up to 30% (e.g., 100 cm$^{-3}$ over Sahara and 200 cm$^{-3}$ over
Gobi and Taklimakan). A notable increase is also calculated over eastern China and
northern India (up to 150 cm$^{-3}$ or 10%) where mineral dust is mixed with
anthropogenic compounds. Remote from the main deserts (e.g., over central Europe),
the change in CDNC is negligible since the contribution of mineral dust particles on
cloud droplet formation is low. Overall, the calculated global average CDNC
increases in the sensitivity simulation by 5% (or 12 cm$^{-3}$).

**5.3 Sensitivity to the emitted dust aerosol load**
The final sensitivity test assumes 50% lower emissions of mineral dust compared
to the reference simulation. The lower tropospheric dust aerosol load in the sensitivity
simulation (49%) results in a 10-30% (up to 150 cm$^{-3}$) decrease of CDNC over the
main deserts. On the other hand, CDNC increases over the anthropogenic (e.g., East
Asia) and biomass burning (e.g., central Africa) regions by 5-10% (up to 150 cm$^{-3}$).
The opposing responses of CDNC to mineral dust emissions result from the fact that
the tropospheric load of the other aerosol species does not change significantly
between the two simulations since the chemical and thermodynamic interactions of



mineral cations with air pollution are still important even after the 50% emission
reduction of dust. Therefore, the presence of inorganic anions (e.g., $NO_3^-$) in the
aerosol phase remains almost unchanged between the two simulations which results in
a decrease of the insoluble fraction of the aerosol, given that mineral dust
concentrations are significantly lower in the reference simulation, leading to higher
CCN activity. Over the Taklimakan desert the insoluble fraction of the aerosol
changes by less than 10%, and therefore, the change in aerosol number concentration
(~40%) due to the mineral dust emission change dominates the effect on CDNC,
which is calculated to be about 100 $cm^{-3}$ (or ~20%) lower in the sensitivity
simulation. On the other hand, over Southeast Asia, the aerosol number concentration
changes less than 10% while the insoluble fraction of the aerosols decreases by 40%.
The significant decrease of $\varepsilon_i$ in Eq. (3) affects the calculated critical supersaturation
of the particle as well as the exponent $x$ in Eq. (4) resulting in an increase of CDNC
by about 150 $cm^{-3}$ (or ~10%). Overall, the impact of halving mineral dust emissions
on the calculated global average CDNC is remarkably small (~3% or 6 $cm^{-3}$).


**6  Summary and Conclusions**
This study assesses the impact of mineral dust on global cloud droplet number
concentrations by using an interactive aerosol-chemistry-cloud-climate model
(EMAC). The "unified dust activation framework" (UAF) has been implemented into
the EMAC model to account for the effects of dust particles through both the
hydrophilicity from adsorption and the acquired hygroscopicity from pollution solutes
(chemical aging) on CCN activity calculations. The calculation of cloud droplet
formation from soluble particles is carried out by using the κ-Köhler theory, while
that of insoluble particles is based on the FHH multilayer adsorption isotherm
approach. For atmospheric particles that contain a substantial fraction of both soluble
(e.g., nitrate) and insoluble material (e.g., mineral dust), cloud formation is calculated
using the UAF, which determines the maximum equilibrium water vapor
supersaturation over an aerosol consisting of an insoluble core with a soluble coating.
Furthermore, the model setup includes thermodynamic interactions between mineral
dust anions (i.e., $Na^+$, $Ca^{2+}$, $K^+$, $Mg^{2+}$) and inorganic cations (i.e., $NO_3^-$, $Cl^-$, $SO_4^{2-}$).
The simulated CDNC at 940 mb, i.e., at cloud base, is relatively high over the
industrialized areas of Europe, Asia and North America (exceeding 1,000 $cm^{-3}$) and



over the biomass burning regions in the tropics (300-700 cm$^{-3}$). Relatively high
CDNC is also calculated over the main deserts (100-1,000 cm$^{-3}$) where the CCN
activity of pristine mineral dust is enhanced by chemical and thermodynamic
interactions with soluble compounds from anthropogenic (including biomass burning)
and natural sources. Low CDNC (around 50 cm$^{-3}$) is calculated over the remote
oceans while CDNC is much higher (up to 1,000 cm$^{-3}$) over more polluted marine
regions near the coast. In view of CDNCs from in situ and satellite observations, we
conclude that the model tends to underestimate CDNC over clean marine areas and
overestimates CDNC over polluted regions.
To estimate the effects of mineral dust and its variable chemical composition on
CDNC, three main sensitivity simulations have been conducted. In the first, mineral
dust emissions were switched off. This reveals that despite the large tropospheric load
of mineral dust aerosols (35 Tg in the base case simulation) the dust presence
decreases the calculated global average CDNC by only 11%. This is the net result of
substantial positive and negative, partly compensating effects. Over polluted regions
(e.g., Europe), dust particles, mostly transported from the Sahara, are mixed with
pollution aerosols resulting in a significant reduction of the CCN activity of the
anthropogenic particles and hence cloud droplet formation. On the other hand, the
activation of freshly emitted dust particles through water adsorption results in an
increase of CDNC over the main deserts. However, on a global scale this does not
match the calculated decrease over the polluted regions. While such sensitivity tests
do not relate to real-world changes, they help understand the role of mineral dust in
the climate system, and especially the importance of including these processes into
climate models, being hitherto neglected.
A second simulation has been performed by switching off the mineral dust
chemistry to estimate the impact of interactions between inorganic and mineral
cations on the predicted CDNC. We find that the tropospheric burden of inorganic
anions (mainly nitrate) increases, resulting in a slight increase of CCN activity and
cloud droplet formation in areas that are influenced by biomass burning and industrial
emissions. Furthermore, including crustal cation chemistry and thermodynamics
significantly affects the aging of mineral dust and its solubility, especially due to the
uptake of nitric acid, so that dust is removed more efficiently through wet and dry
deposition. This results in a decrease of CDNC over the remote deserts (e.g.,





Taklimakan). On average, global CDNC does not change significantly by considering
mineral dust chemistry and thermodynamics.
In the third simulation the FHH calculations have been switched off to estimate the
effects of water adsorption onto the surface of insoluble dust particles on the predicted
CDNC. The CDNC in the reference simulation is found to be higher over arid areas
due to the adsorption activation of the freshly emitted insoluble dust particles. On the
other hand, CDNC is lower over polluted regions (e.g., over Europe) since the aged
dust particles experience significant water uptake during their activation reducing the
$s_{max}$ and the activation of the smaller anthropogenic particles. Overall, the use of the
UAF results in a decrease of the global average CDNC by ~10%. This result shows
that for the modeling of cloud droplet formation, adsorption activation of insoluble
aerosols is more important than mineral dust chemistry and thermodynamics.
However, taking into account the adsorption activation of insoluble aerosols without
mineral dust chemistry can result in a significant overestimation of CDNC, mainly
over the remote deserts. Conversely, considering mineral dust chemistry and
thermodynamics without UAF can result in significant overestimation of CDNC over
polluted areas.
Finally, three additional sensitivity simulations have been conducted to investigate
the sensitivity of the results to the physicochemical properties of the emitted mineral
dust (chemical composition, hydrophilicity and emission strength). This indicates that
the calculated CDNC is sensitive to the mineral dust hydrophilicity and emission load.
By assuming drastic differences in the dust source and the dust hydrophilicity, we find
only small (~5%) changes in the average CDNC. Further, the global average CDNC is
not sensitive to the chemical composition of mineral dust.
This study demonstrates that a comprehensive treatment of the CCN activity of
mineral dust aerosols and their chemical and thermodynamic interactions with
inorganic species by CCMs is important to realistically account for aerosol-chemistry-
cloud-climate interactions. Neglecting the adsorption activation of freshly emitted
dust can result in significant biases over areas close to deserts. In addition, neglecting
the mineral dust chemistry and thermodynamics results in an underestimation of the
coating of dust by hygroscopic salts during atmospheric aging. The realistic
representation of soluble coating on dust is crucial since it affects its efficiency to
grow by water uptake, which significantly influences the local supersaturation and



thus cloud droplet formation over anthropogenically polluted regions. In this first
study we apply the UAF diagnostically, while for future applications, e.g., to simulate
climate effects, we plan prognostic climate calculations where effects on precipitation
formation and dynamical responses will also be accounted for.

**Acknowledgements**
V.A. Karydis acknowledges support from a FP7 Marie Curie Career Integration
Grant (project reference 618349). A.P. Tsimpidi acknowledges support from a DFG
individual grand program (project reference TS 335/2-1).






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



**Table 1.** Comparison of simulated and observed (Karydis et. al, 2011 and the references therein) cloud droplet number concentrations

| Location | Lat. | Long. | Alt. | Time | Observation | Simulation |
|---|---|---|---|---|---|---|
| S. Pacific Ocean | | | PBL | Annual | 40 | 23 |
| S. Pacific Ocean | 20S-35S | 135W-175W | PBL | Annual | 82 | 26 |
| Eastern Pacific Ocean | 29N-32N | 120W-123W | 450-850m | July | 49-279 | 133 |
| N. Pacific Ocean | 41N | 131W | <1500m | April | 21-74 | 51 |
| N. Pacific Ocean | | | PBL | Annual | 64 | 59 |
| W. of Canary Islands | 32N | 25W | PBL | July | 17 | 115 |
| N. Atlantic Ocean | | | PBL | Annual | 89 | 112 |
| S. Atlantic Ocean | | | PBL | Annual | 67 | 51 |
| S. Indian Ocean | | | PBL | Annual | 42 | 29 |
| West Australia (remote) | 30S-40S | 88E-103E | PBL | Annual | 107 | 22 |
| Beaufort Sea (Western Arctic Ocean) | 72N-78N | 154W-159W | 202-1017m | June | 178-365 | 25 |
| Beaufort Sea (Western Arctic Ocean) | 70.5N-73N | 145N-147N | 300-3000m | June | 20-225 | 28 |
| Beaufort Sea (Western Arctic Ocean) | 65N-75N | 130W-170W | 400-4600m | April | 48-77 | 39 |
| Northeast Alaska coast | 69N-71N | 150W-158W | 400-4000m | October | 10-30 | 23 |
| Yellow Sea (Eastern coast of China) | 28N-31N | 127E-131E | PBL | Annual | 30-1000 | 764 |
| SE Asia coast | 10N-40N | 105E-150E | PBL | Annual | 186 (100-250) | 522 |
| NE Asia coast | | | PBL | Annual | 129 | 768 |
| N. America coast (Pacific) | | | PBL | Annual | 96 | 91 |
| N. America coast (Pacific) | 15N-35N | 115W-140W | PBL | Annual | 159 (150-300) | 190 |
| S. America coast (Pacific) | | | PBL | Annual | 77 | 75 |
| S. America coast (Pacific) | 8S-28S | 70W-90W | PBL | Annual | 182 (100-300) | 186 |
| N. Africa coast (Atlantic) | | | PBL | Annual | 95 | 123 |
| S. Africa coast (Atlantic) | | | PBL | Annual | 95 | 107 |
| S. Africa coast (Atlantic) | 5S-25S | 10W-15E | PBL | Annual | 153 (130-300) | 189 |
| Eastern N. Atlantic Ocean | 50N-55N | 25W-30W | 800-2200m | April | 65-300 | 39 |
| NW coast of Santa Maria, Azores | 37N | 25W | 550-1000m | June | 150 (74-192) | 83 |
| Canary Islands Vicinity | 28N | 16.5W | PBL | June-July | 51-256 | 174 |
| Canary Islands Vicinity | 28N | 16.5W | PBL | June-July | 90-300 | 174 |
| Atlantic Ocean (W. of Morocco) | 34N | 11W | PBL | July | 77 | 114 |
| Coast of Oregon | 45.5N | 124.5W | PBL | August | 25-210 | 124 |
| Key West, FL | 24.5N | 82W | PBL | July | 268-560 | 318 |
| Bay of Fundy, Nova Scotia, Canada | 44N | 66W | 20-290m | August | 61 (59-97) | 246 |
| Cornwall Coast (SW UK) | 50N | 5.5W | 450-800m | February | 130 | 602 |
| British Isles, UK | 55N | 2.5W | Surface | April | 172 | 287 |
| British Isles, UK | 51N | 6W | Surface | October | 119 | 71 |
| British Isles, UK | 53N | 9.5W | Surface | December | 96 | 318 |
| SE coast of England | 51.5N-52N | 1.5E-2.5E | 380-750m | September | 151-249 | 1019 |
| Indian Ocean (SW of India) | 10S-10N | 65E-75E | 50-550m | February-March | 100-500 | 520 |



| Location | Lat. | Long. | Alt. | Time | Observation | Simulation |
|---|---|---|---|---|---|---|
| Qinghai Province (Western China) | 34N-37N | 98E-103E | PBL | Annual | 30-700 | 585 |
| Beijing, China | 37N-41N | 113E-120E | PBL | Annual | 30-1100 | 1185 |
| NE China (East of Beijing) | 39N-40N | 117.5E-118.5E | 1719-1931m | April-May | 200-800 | 813 |
| Hebei Province (Central Eastern China) | 35N-40N | 112E-119E | PBL | Annual | 30-400 | 1150 |
| Cumbria, N. England | 54.5N | 2.5W | Surface | March-April | 100-2000 | 743 |
| Cumbria, N. England | 54.5N | 2.5W | Surface | May | 482-549 | 840 |
| Koblenz, Germany | 50N | 7.5E | 901-914hPa | May | 675-900 | 1258 |
| Koblenz, Germany | 50N | 7.5E | 945hPa | October | 965 | 1039 |
| Northern Finland | 68N | 24E | 342-572m | Annual | 154 (30-610) | 332 |
| Kuopio, Finland | 62.5N | 27.5E | 306m | August-November | 138 | 1142 |
| Northern Finland | 68N | 24E | 342-572m | October-November | 55-470 | 336 |
| Cabauw, Netherland | 51N | 4.5E | PBL | May | 180-360 | 946 |
| Jungfraujoch, Switzerland | 46.5N | 7.5E | Surface | July-August | 112-416 | 176 |
| Barrow, AK | 71.5N | 156.5W | 389-830m | August | 56 | 47 |
| Barrow, AK | 71.5N | 156.5W | 431-736m | May | 222 | 26 |
| Barrow, AK | 71.5N | 156.5W | 297-591m | June | 121 | 31 |
| Barrow, AK | 71.5N | 156.5W | 393-762m | July | 54 | 29 |
| Barrow, AK | 71.5N | 156.5W | 1059-1608m | September | 81 | 23 |
| Southern Great Plains, OK | 36.5N | 97.5W | 795-1450m | Winter | 265-281 | 341 |
| Southern Great Plains, OK | 36.5N | 97.5W | 343-1241m | Winter | 244 | 341 |
| Southern Great Plains, OK | 36.5N | 97.5W | 985-1885m | Spring | 200-219 | 384 |
| Southern Great Plains, OK | 36.5N | 97.5W | 671-1475m | Spring | 203 | 537 |
| Southern Great Plains, OK | 36.5N | 97.5W | 1280-2200m | Summer | 128-159 | 393 |
| Southern Great Plains, OK | 36.5N | 97.5W | 756-1751m | Summer | 131 | 603 |
| Southern Great Plains, OK | 36.5N | 97.5W | 1030-1770m | Autumn | 217-249 | 505 |
| Southern Great Plains, OK | 36.5N | 97.5W | 404-1183m | Autumn | 276 | 642 |
| Southern Great Plains, OK | 36.5N | 97.5W | 900-800hPa | March | 200 (100-320) | 563 |
| Southern Great Plains, OK | 36.5N | 97.5W | 300-600m | April | 650 | 1159 |
| Southern Great Plains, OK | 36.5N | 97.5W | 700-1200m | September-October | 457 | 740 |
| Cleveland, OH; Detroit, MI | 40N-42.5N | 80.5W-85W | 300-1000m | August | 320-1300 | 817 |
| Central Ontario, Canada | 50N | 85W | <2500m | October | 147 (119-173) | 201 |
| Central Ontario | 50N | 85W | 2000-2100m | Summer | 350-360 | 143 |
| Central Ontario | 50N | 85W | 1300m | Winter | 190 | 112 |
| Upper NY State | 44N | 75W | 1500m | Autumn | 240 | 583 |
| State College, Pennsylvania | 41N | 78W | 1000-1600m | October | 388 | 551 |
| Mount Gibbes, NC | 35.5N | 82W | Surface | Annual | 238-754 | 392 |
| Cape Kennedy, FL | 28.5N | 80.5W | 600-2800m | August | 250-330 | 134 |



**Table 2.** Statistical evaluation of EMAC CDNC against 74 worldwide observational datasets derived from in situ measurements and satellite retrievals.

| Site Type | Number of datasets | Mean Observed (cm⁻³) | Mean Simulated (cm⁻³) | MAGE (cm⁻³) | MB (cm⁻³) | NME (%) | NMB (%) | RMSE (cm⁻³) |
|---|---|---|---|---|---|---|---|---|
| Clean marine | 14 | 86 | 53 | 51 | -33 | 60 | -39 | 81 |
| Polluted marine | 24 | 169 | 296 | 159 | 127 | 94 | 75 | 263 |
| Continental | 37 | 339 | 536 | 269 | 198 | 80 | 58 | 358 |
| **Total** | 75 | 237 | 369 | 193 | 132 | 82 | 56 | 295 |



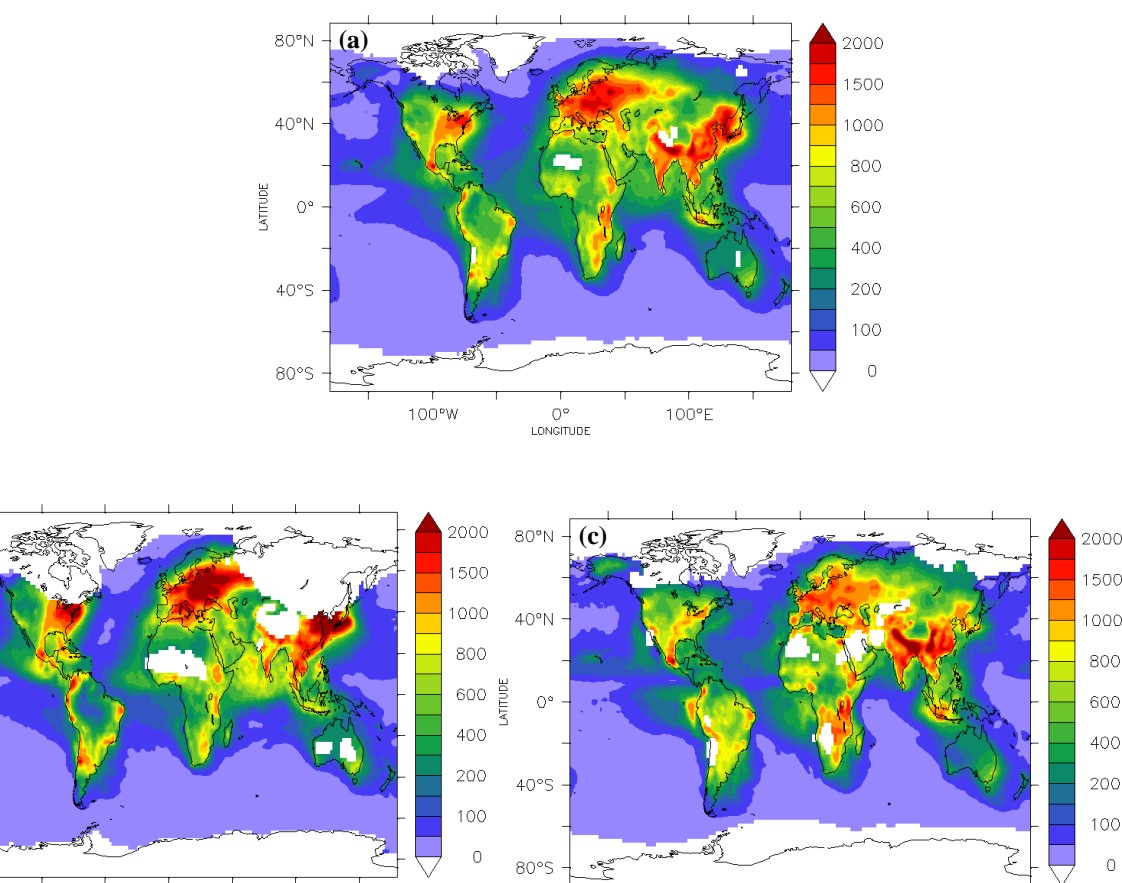

**Figure 1**: Predicted **(a)** annual, **(b)** DJF, and **(c)** JJA mean cloud droplet number concentrations (cm$^{-3}$) at the lowest cloud-forming level (940 mb). White color represents areas that are cloud-free or covered by ice clouds.



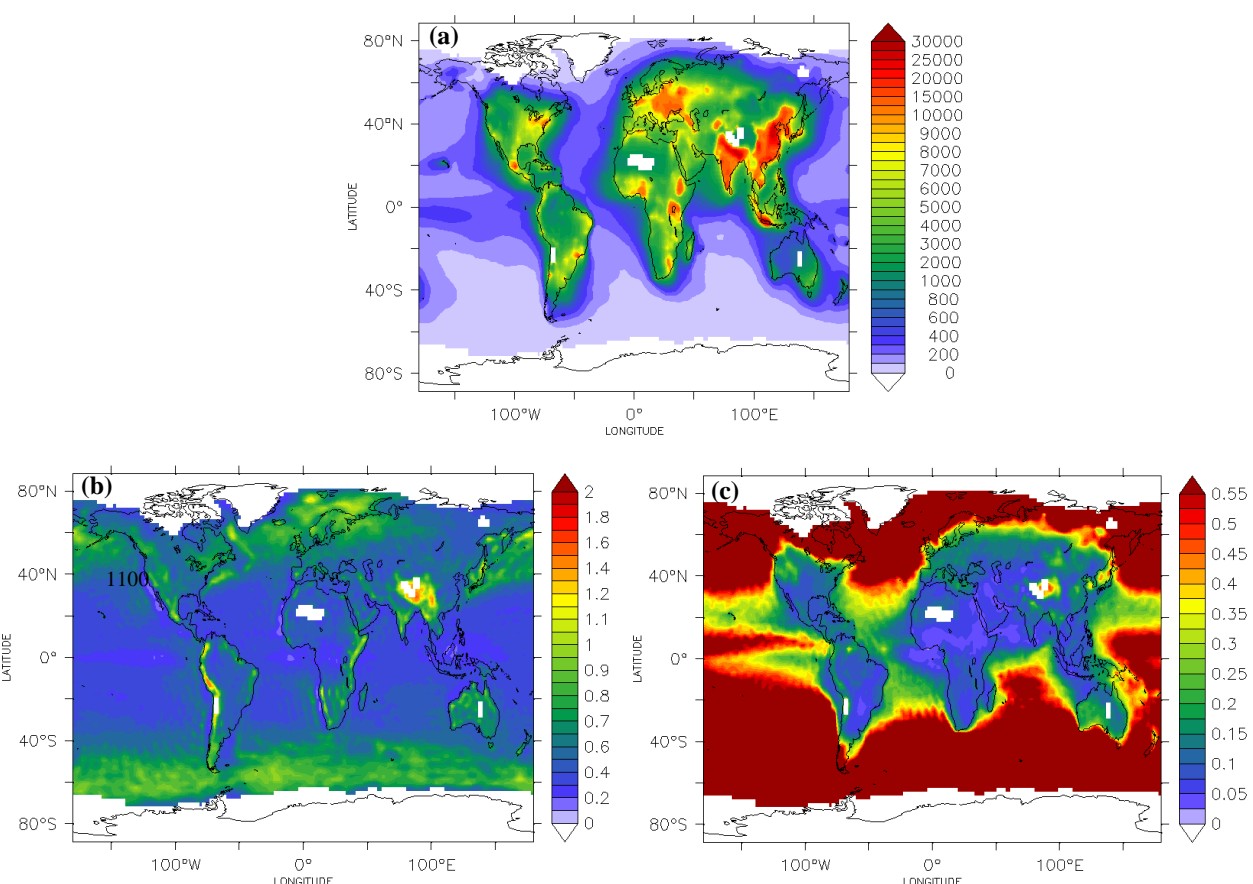

**Figure 2**: Predicted annual mean **(a)** aerosol number concentration (cm$^{-3}$), **(b)** large-scale cloud updraft velocity (m s$^{-1}$), and **(c)** maximum supersaturation (%) at the lowest cloud-forming level (940 mb). White areas correspond to regions where liquid cloud droplets do not form.





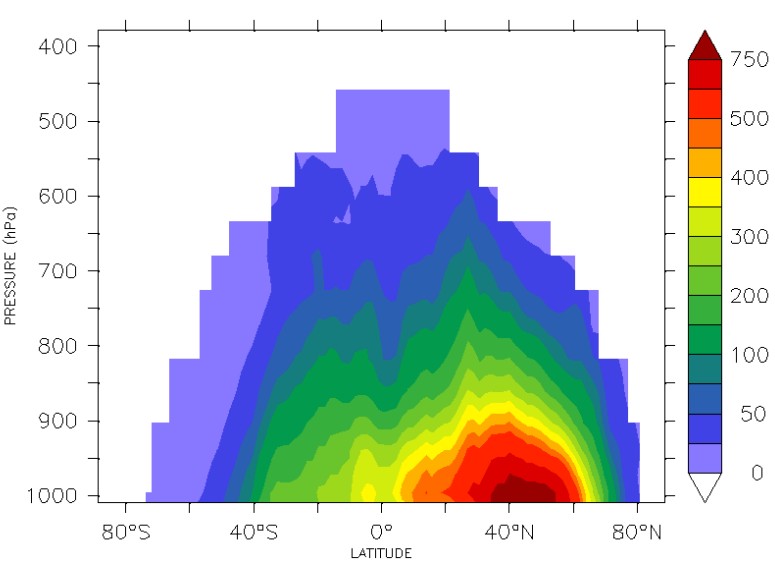

**Figure 3:** Predicted zonal annual mean cloud droplet number concentration (cm⁻³). White areas correspond to regions where liquid cloud droplets do not form.





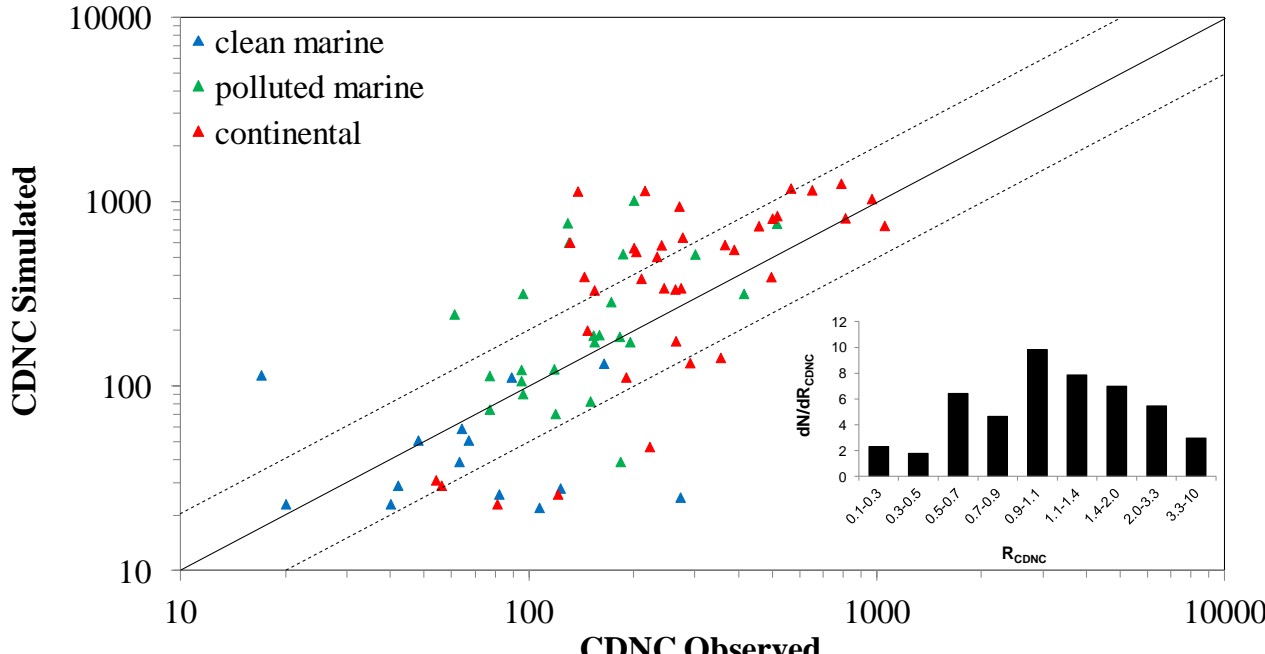

**Figure 4:** Scatterplot comparing model simulated cloud droplet number concentrations (cm$^{-3}$) against 74 worldwide observational datasets derived from in situ measurements and satellite retrievals. Also shown are the 1:1, 2:1, 1:2 lines, and the probability distribution of the ratio of the simulated CDNC to the observed CDNC (R$_{CDNC}$), where N is the number of occurrences in each R$_{CDNC}$ (inset plot).





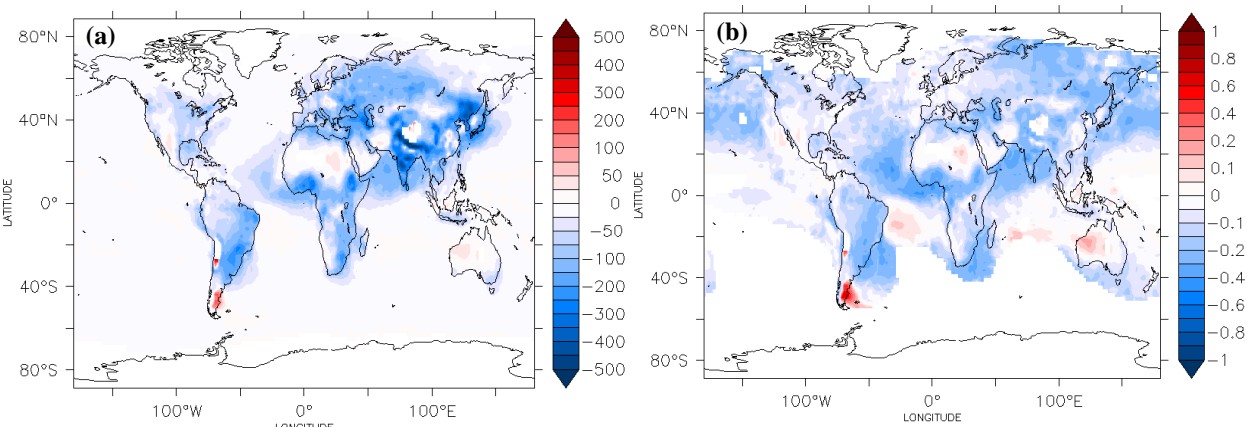

**Figure 5: (a)** Absolute (in $cm^{-3}$) and **(b)** fractional annual change of the predicted CDNC (at the lowest cloud-forming level, 940 mb) by switching on/off the mineral dust emissions. A positive change corresponds to an increase from the presence of dust.



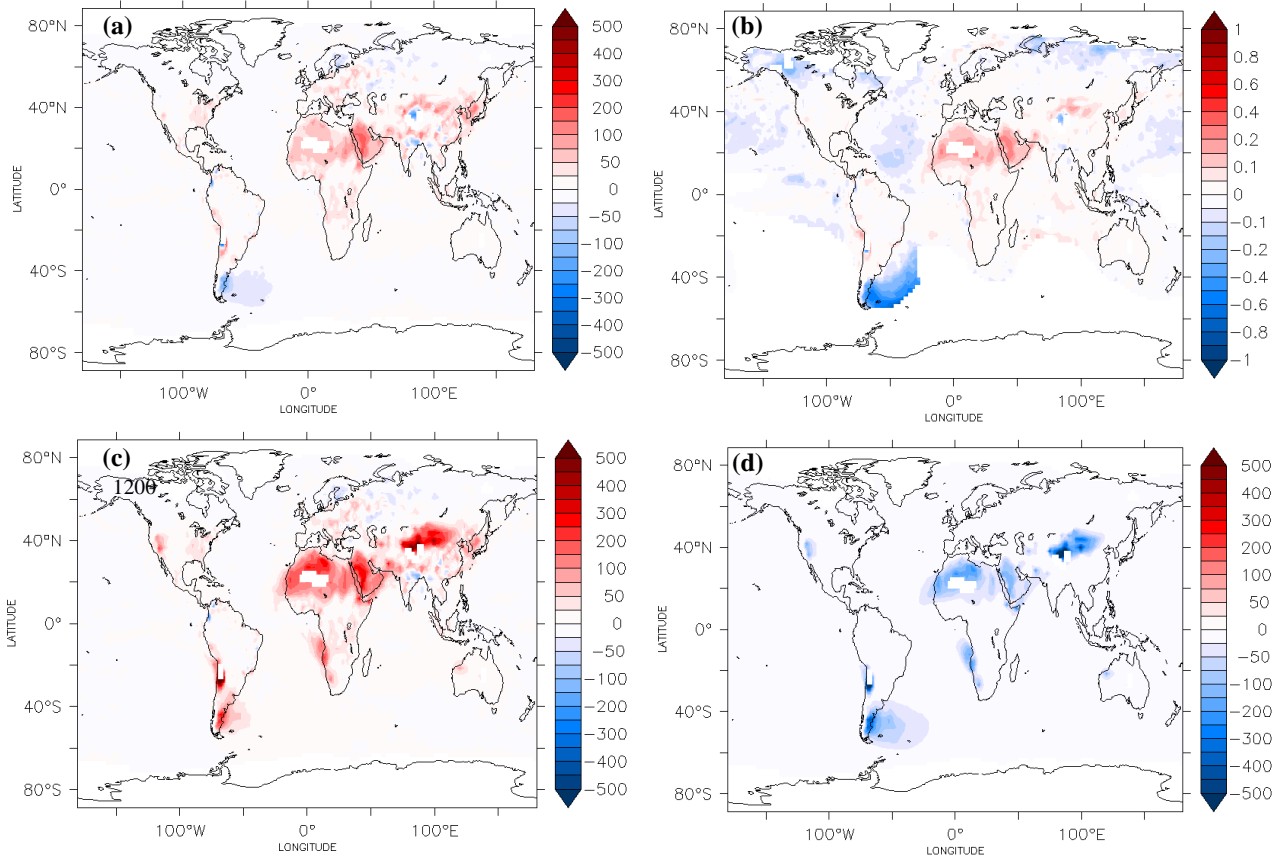

**Figure 6:** **(a)** Absolute (in cm⁻³) and **(b)** fractional annual average change of the predicted total CDNC, and absolute (in cm⁻³) annual average change of the CDNC from **(c)** soluble and **(d)** insoluble particle modes, by switching on/off the mineral dust chemistry. Concentrations reported at the lowest cloud-forming level (940 mb). A positive change corresponds to an increase from dust–chemistry interactions.



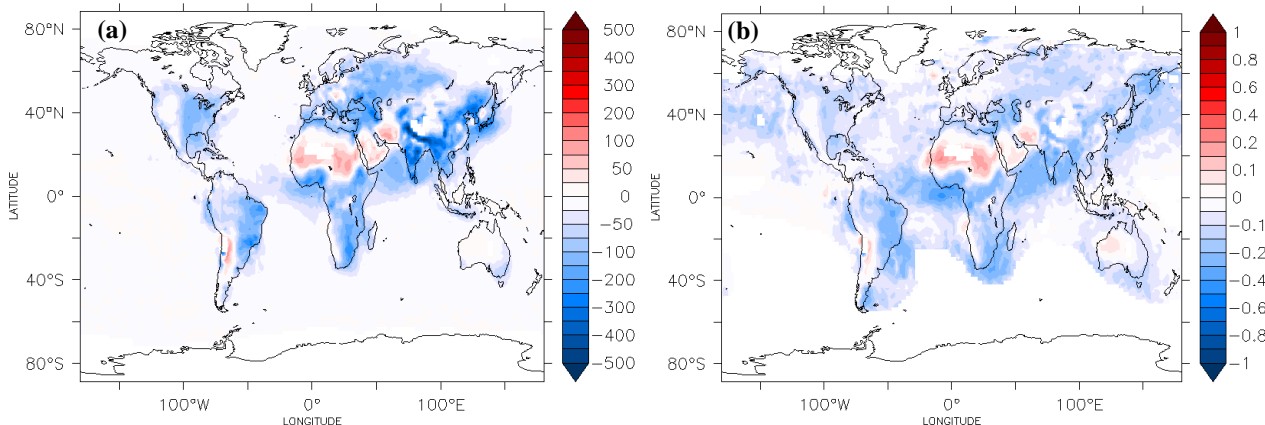

**Figure 7:** **(a)** Absolute (in cm$^{-3}$) and **(b)** fractional annual average change of the predicted CDNC (at the lowest cloud-forming level, 940 mb) by switching on/off the FHH adsorption activation physics. A positive change corresponds to an increase from water adsorption on mineral dust.





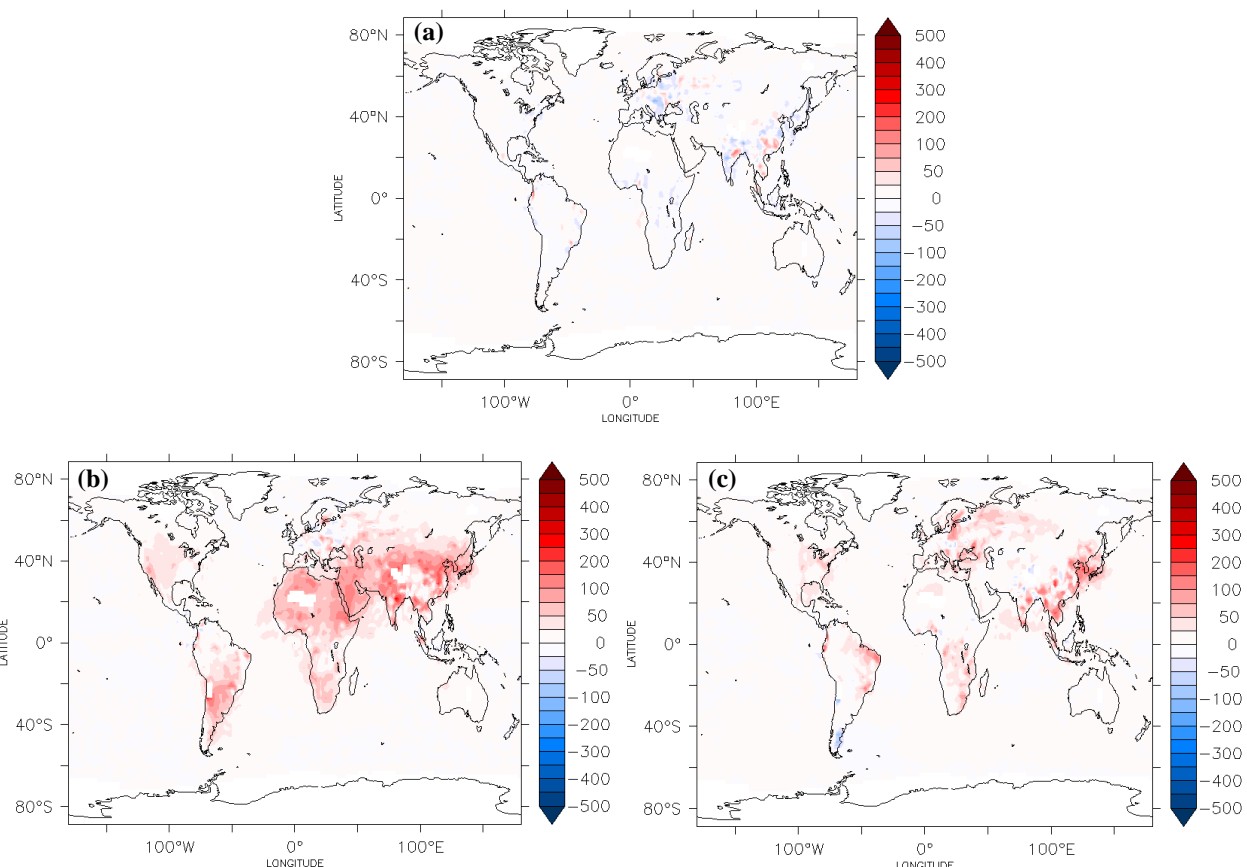

**Figure 8:** Absolute changes (in cm$^{-3}$) of the predicted annual average CDNC by **(a)** assuming a globally uniform chemical composition of mineral dust, **(b)** increasing the $B_{FHH}$ hydrophilicity parameter of dust by 10%, and **(c)** reducing mineral dust emissions by 50%. A positive change corresponds to an increase relative to the reference simulation.