# Peer review of "Global impact of mineral dust on cloud droplet number concentration"

_Atmospheric Chemistry and Physics, 2016_

## Referee Comment (RC1) · Anonymous Referee #1 · 29 Dec 2016

This is an interesting study of the role of dust in droplet nucleation. Although some of the conclusions are compromised by neglect of droplet collision, I don't think those concerns need to be addressed in this study. There might even be value in neglecting droplet collisions, although that raises questions about the evaluation.

Minor comments

Lines 65-58. Confusing text. I suggest instead "Reports of hygroscopic growth
measurements of dust particles indicate solubility to be very low, so that activation of observed cloud condensation nuclei (CCN) has been attributed to soluble ions
present in the particles".

Line 71. Wouldn't the "fraction of soluble material on the particles" correspond to the soluble ions referred to above?
Or is the critical distinction between soluble material

within and on the surface of the particles? And between the fresh dust and aged dust? Perhaps even fresh dust is coated with soluble ions. This is not to say that adsorption or condensation of secondary soluble material are not important, but why neglect soluble material in the emitted dust. Surely some types of dust (clays?) must contain soluble material.

Line 90. Start new paragraph with "Hatch".

Line 96. Start new paragraph with "Based".

Lines 117. Start new paragraph with "Soluble", as the previous text describes mechanism while the following text describes conclusions above dust activity sampled in the atmosphere.

Line 143. Drop "Only", as "few" implies it.

Line 159. I think you mean "aged dust can substantially deplete in-cloud supersaturation ", and replace "eventually" with "hence".

Line 179. Replace "which " with "that".

Line 201. Replace "is" with "are".

Section 3.1. This discussion never mentions the role of droplet collision in depleting droplet number concentration. Droplet activation is not the only process that determines droplet number concentration. Please consider the role of collision in your discussion, or show that it is not important (perhaps in thin warm clouds).

Line 336. Are these in-cloud means?

Line 338. Replace "are" with "is".

Line 381. The grid cell mean is typically less than 1 cm/s in global models. How large is the mean velocity over the central Asian deserts?

Line 384. Is 113 the global annual mean?
**Interactive comment**

Section 3.2 I'm not sure what the purpose of this section is, since the aerosol and updraft velocity are not evaluated. Are you trying to show that the activation process is realistic, or just that droplet numbers are realistic? I'm not sure that you can achieve the former without validating the aerosol and updraft velocity too (or stratifying droplet number by aerosol and updraft velocity), and the latter is of limited value because EMAC neglects collision (as we learn later).

Line 408. In-cloud values?

Line 441. Spatial and/or temporal variability?

Line 468. Now we finally learn that collision is neglected in the simulations. This should be noted before the comparisons are presented.

Line 493. This gets confusing. Please be explicit about whether you are referring to addition or subtraction of mineral dust.

Line 502-503. This is the first time we learn about nudging. This should be reported in the experiment design.

Section 4.2. This is written very clearly and is quite interesting.

Line 607. Over over.

Lines 665-667. Should note again that the simulation neglects droplet collision. . . . .

---

## Referee Comment (RC2) · Anonymous Referee #2 · 1 Feb 2017

Anonymous Reviewer #2. Review of manuscript "Global impact of mineral dust on cloud droplet number concentration" by V. Karydis et al.

This work uses a suite of models (including the atmospheric chemistry model ECHAM5/MESSy, MECCA, aerosol thermodynamics with ISORROPIA-II, and a series of other aerosol micro-physics subroutines) in order to explore, through numerical experiments, the potential global impact of wind-blown mineral dust in the number concentration of activated cloud droplets. Three mechanisms are explored in the paper: adsorption over insoluble dust particles, classical activation on particles with soluble coating, and a second order effect which involves interaction of the mineral cations in the dust particles with other inorganic aerosols. These mechanism are explored through sensitivity simulations in which the model is run with/without the process under consideration. The paper is relevant and well written. The material presented is

novel since it attempts to quantify potential impacts of mechanism not previously considered. However, I think the discussion of the implications of these mechanism should be performed in much more depth than what is done in the paper, and some substantial modifications in technical details are needed for the paper to be published. The difference between the conclusions found in this study and a previous work (Karydis et. al 2011) should be made explicit.

General Comments:

- No description of the cloud scheme utilized in the model is done. Therefore, it is not clear under which conditions is the activation parameterization triggered. Very little or no mention of cloud microphysics is done in the paper. The distribution of low level cloudiness in the model is not presented, which would be crucial to determine the actual extent of global impact of CDNC on aerosol-cloud-radiation interactions.

- Although the paper is mainly focused on the impacts of dust on CDNC, no mention is done regarding the impact of dust on number concentration of aerosol particles that could activate. It would help in the interpretation of the results to know what the impact of switching dust emissions off is on the number and size of aerosol particles. A figure showing the changes caused by dust on the aerosol particles should be shown next to Figure 5.

- There is no mention in the paper about the geographic distribution of soluble and insoluble fractions in the dust modes predicted by the model. This would definitely help with the discussion and interpretation of the results. A map showing this distribution would help understanding the underlying processes.

- The paper does not explain how the CDNC shown in the maps is calculated. Are those grid-cell averages? Are those in-cloud values? Is this the value only after activation subroutine is called? Or are these values produced by the full cloud-microphysical scheme?

- No indication of the frequency of occurrence of liquid clouds at the level in the model is mentioned, nor that of the climatological cloud cover in those regions. If this is somehow included in the manuscript, the overall importance of dust on CDNC globally could be better assessed. The specifics of the annual average CDNC shown in the paper should be discussed and described in detail.

- Some fundamental issues with the unified theory should be discussed by the authors in this manuscript. In particular the potential oversimplification of the activation process for insoluble particles with small soluble coatings (as could potentially be the case for dust particles). See specific comments.

- I suggest modifying some of the conclusions of the paper, since they can be over-reaching. It doesn't seem that the paper actually "demonstrates" that the biases are substantial, or that this treatment is indeed correct. In fact, the authors acknowledge almost no sensitivity of CDNC to massive cuts in dust emissions, or hydrophilicity parameter, or on dust chemical composition. For example, I quote "By assuming drastic differences in the dust source and the dust hydrophilicity we find only small ($\sim$5%) changes in the average CDNC".

Specific Comments:

Section 2.2. Line 265. It is not clear from the equations nor the references cited in the document, how can an exponent $x = -3/2$ be obtained from equation (3) when there is insoluble material but no FHH terms. The -3/2 exponent arises from the fact that the whole volume of the particle contributes to the soluble material during the activation process. It is not explicit from the document what is the expression relating critical diameter and critical supersaturation when there is a substantial fraction of insoluble material (i.e., in equation 3, with no FHH terms, but a small amount of soluble material). The relations between dry aerosol size and critical supersaturation are severely modified when an insoluble core is present (see for example, Pruppacher and Klett, chapter 6, equations 6-37 to 6-42). Therefore, there is a possibility that one could see substantial changes in CDNC by simply improving the description of the relation between critical diameter and critical supersaturation for cases where there is an insoluble core (no FHH terms). This issue should be explored and discussed in the paper.

- It would be convenient for the readers to see average values of CDNC, or average fractional changes printed in the global maps of figures 1, 2, 5, 6, 7 and 8.

- The difference between results shown in Figure 5, and Figure 8c are not entirely clear to me. So figure 5 has no mineral dust emissions, and Figure 8c, was performed with 50% aerosol emissions compared to base case? So in the case of no emissions, there is a net decrease in CDNC, but when there is only a 50% decrease in the emission load there is an increase in CDNC?

- Line 712. Should it read "insensitive"?

- Figure 2. Are these values grid-cell averages? Or are they in-cloud values only?

- From figure 8, it seems that BFHH parameter has a larger (or at least comparable) impact to reducing mineral dust emissions by 50%? This should be discussed in much more detail. As mentioned above, perhaps showing the net impact that the 50% reduction in dust emissions has on aerosol number concentration would be helpful in the interpretation of the results.

- Similarly, the paper shows very little sensitivity of CDNC to dust chemical composition, but relatively high sensitivity to the BFHH parameter. However, it is reasonable to believe that the FHH parameters are linked to the chemical composition of the mineral dust particles. Therefore, some discussion should be included regarding the relationship between the FHH theory parameters and dust chemical composition, and the potential impacts it could have in the simulations.
* * *

---

## Author Comment (AC1) · 24 Mar 2017

> *This is an interesting study of the role of dust in droplet nucleation. Although some of the conclusions are compromised by neglect of droplet collision, I don't think those concerns need to be addressed in this study. There might even be value in neglecting droplet collisions, although that raises questions about the evaluation.*

We would like to thank the reviewer for his/her positive response. Indeed, as discussed in the text, the CDNC shown in this study is equal to the nucleated droplet number concentration before the collision and coalescence processes, which we acknowledge as an upper limit in clouds. This may result on the overestimation of CDNC in some areas. However, over polluted regions, where the model overestimates CDNC, the sensitivity of cloud albedo ($R_c$) to CDNC is low. For typical values of cloud albedo ($0.28 \leq Rc \leq 0.72$) $\Delta R_c = 0.075 \Delta \ln(\text{CDNC})$ (Seinfeld and Pandis, 2006). Therefore, Cloud albedo sensitivity to CDNC decreases with increasing CDNC. Based on the typical properties of stratus clouds, a 30% overestimation of CDNC results in 2.25% increase to cloud albedo and in a perturbation of -1.1 W m$^{-2}$ in the global mean cloud radiative forcing (Schwartz, 1996). Below is a point by point response to the reviewer's comments.

**Minor comments**

1. *Lines 65-58. Confusing text. I suggest instead "Reports of hygroscopic growth measurements of dust particles indicate solubility to be very low, so that activation of observed cloud condensation nuclei (CCN) has been attributed to soluble ions present in the particles".*

   We adopted the reviewer's suggestion and changed the text accordingly.

2. *Line 71. Wouldn't the "fraction of soluble material on the particles" correspond to the soluble ions referred to above? Or is the critical distinction between soluble material within and on the surface of the particles? And between the fresh dust and aged dust? Perhaps even fresh dust is coated with soluble ions. This is not to say that adsorption or condensation of secondary soluble material are not important, but why neglect soluble material in the emitted dust. Surely some types of dust (clays?) must contain soluble material.*

   Yes, in Line 71, the "fraction of soluble material on the particles" corresponds to the soluble ions referred in line 66. In this study we have implicitly taken into account the presence of soluble material in the freshly emitted dust by assuming that the emitted mineral particles are a mixture of inert material (i.e., bulk dust) with reactive components (i.e., $Ca^{2+}$, $Mg^{2+}$, $K^+$, and $Na^+$) that form soluble salts.

3. *Line 90. Start new paragraph with "Hatch".*

   Done.

4. *Line 96. Start new paragraph with "Based".*

Done.

5. *Lines 117. Start new paragraph with "Soluble", as the previous text describes mechanism while the following text describes conclusions above dust activity sampled in the atmosphere.*

Done.

6. *Line 143. Drop "Only", as "few" implies it.*

Corrected.

7. *Line 159. I think you mean "aged dust can substantially deplete in-cloud supersaturation ", and replace "eventually" with "hence".*

Corrected.

8. *Line 179. Replace "which " with "that".*

Done.

9. *Line 201. Replace "is" with "are".*

Done.

10. *Section 3.1. This discussion never mentions the role of droplet collision in depleting droplet number concentration. Droplet activation is not the only process that determines droplet number concentration. Please consider the role of collision in your discussion, or show that it is not important (perhaps in thin warm clouds).*

In this study, droplet depletion by collision, coalescence and collection are not taken into account. Therefore, CDNC values presented in this section can be considered as an upper limit. This is now pointed out at the beginning of the section.

11. *Line 336. Are these in-cloud means?*

Yes, in this study, CDNC is referred to the number concentration of droplets nucleated in-cloud. We added this information in the text.

*12. Line 338. Replace "are" with "is".*

      Corrected.

*13. Line 381. The grid cell mean is typically less than 1 cm/s in global models. How large is the mean velocity over the central Asian deserts?*

   The large scale updraft velocity over the central Asian deserts (e.g., over Gobi) ranges from -0.4 cm s$^{-1}$ to 0.3 cm s$^{-1}$ throughout the year with an annual mean value of 0.01 cm s$^{-1}$.

*14. Line 384. Is 113 the global annual mean?*

      Yes, it is the annual mean over all oceans.

*15. Section 3.2 I'm not sure what the purpose of this section is, since the aerosol and updraft velocity are not evaluated. Are you trying to show that the activation process is realistic, or just that droplet numbers are realistic? I'm not sure that you can achieve the former without validating the aerosol and updraft velocity too (or stratifying droplet number by aerosol and updraft velocity), and the latter is of limited value because EMAC neglects collision (as we learn later).*

   Aerosol fields produced by EMAC have been evaluated against in-situ observations in previous studies (Pozzer et al., 2012; Tsimpidi et al., 2014; Karydis et al., 2016). The cloud droplet formation parameterization used in this work has been also extensively evaluated by comparing computations of CDNC and $S_{max}$ and their sensitivity to aerosol properties against detailed numerical simulations of the activation process by a parcel-model (Betancourt and Nenes, 2014a). Furthermore, the cloud-averaged CDNC for stratocumulus clouds, which are described by EMAC, is well captured by the cloud droplet formation parameterization used in this study (Morales et al., 2011). Considering the influence of droplet collision and coalescence processes may, in part, reduce CDNC prediction biases, however, these processes are becoming important in the presence of clouds with substantial amount of drizzle. The purpose of this section is actually to provide a qualitative evaluation of the model's ability to capture the spatial and temporal variations of CDNC. The model is able to reproduce the increasing CDNC in air masses from clean marine regions to polluted marine and continental regions, though are biased somewhat high over the latter. However, a quantitative evaluation

of the model is not currently feasible since the observations span over a decade (in contrast to the simulation which represents one year) and typically do not represent monthly means over 1.9° grid squares (as sampled from the model results). Furthermore, the model tendency to overestimate the high values of CDNC has small impact on the overall cloud radiative forcing since cloud albedo sensitivity to CDNC decreases with increasing CDNC. Part of this discussion has been added in the revised manuscript.

*16. Line 408. In-cloud values?*

Yes, in this study, CDNC is referred to the number concentration of droplets nucleated in-cloud. We added this information in the text

*17. Line 441. Spatial and/or temporal variability?*

Here we refer to spatial variability. We have now clarified this in the text.

*18. Line 468. Now we finally learn that collision is neglected in the simulations. This should be noted before the comparisons are presented.*

In the revised manuscript, we have also included this information at the beginning of section 3.1.

*19. Line 493. This gets confusing. Please be explicit about whether you are referring to addition or subtraction of mineral dust.*

We refer to changes caused by the addition of mineral dust particles. This is now explicitly stated in the sentence.

*20. Line 502-503. This is the first time we learn about nudging. This should be reported in the experiment design.*

We included this information in section 2.1.

*21. Section 4.2. This is written very clearly and is quite interesting.*

We thank the reviewer for his/her positive comment.

*22. Line 607. Over over.*

Corrected.

23. *Lines 665-667. Should note again that the simulation neglects droplet collision.*

We noted again that we have neglected the collision and coalescence processes, which can lead to an overestimation of CDNC.

---

## Author Comment (AC2) · 24 Mar 2017

*This work uses a suite of models (including the atmospheric chemistry model ECHAM5/MESSy, MECCA, aerosol thermodynamics with ISORROPIA-II, and a series of other aerosol micro-physics subroutines) in order to explore, through numerical experiments, the potential global impact of wind-blown mineral dust in the number concentration of activated cloud droplets. Three mechanisms are explored in the paper: adsorption over insoluble dust particles, classical activation on particles with soluble coating, and a second order effect which involves interaction of the mineral cations in the dust particles with other inorganic aerosols. These mechanism are explored through sensitivity simulations in which the model is run with/without the process under consideration. The paper is relevant and well written. The material presented is novel since it attempts to quantify potential impacts of mechanism not previously considered. However, I think the discussion of the implications of these mechanism should be performed in much more depth than what is done in the paper, and some substantial modifications in technical details are needed for the paper to be published. The difference between the conclusions found in this study and a previous work (Karydis et. al 2011) should be made explicit.*

We thank the referee for the thoughtful review. Below are our responses to the issues raised.

**General comments**

1. *No description of the cloud scheme utilized in the model is done. Therefore, it is not clear under which conditions is the activation parameterization triggered. Very little or no mention of cloud microphysics is done in the paper. The distribution of low level cloudiness in the model is not presented, which would be crucial to determine the actual extent of global impact of CDNC on aerosol-cloud-radiation interactions.*

   The cloud scheme used in this study contains the original cloud process and cover routines from ECHAM5 and calculates the cloud microphysics by using the detailed two-moment liquid and ice-cloud microphysical scheme described in Lohmann and Ferrachat (2010), which enables a physically based treatment of aerosol–cloud interactions. This information has been added in section 2.1. The cloud droplet formation parameterization described in section 2.2 is only triggered when warm clouds are present (i.e., cloud water is present and temperature exceeds 269 K). We have also included this information in the revised manuscript. The distribution of the calculated low-level cloudiness has been added in Figure 2.

2. *Although the paper is mainly focused on the impacts of dust on CDNC, no mention is done regarding the impact of dust on number concentration of aerosol particles that could activate. It would help in the interpretation of the results to know what the impact of switching dust emissions off is on the number and size of aerosol particles. A figure showing the changes caused by dust on the aerosol particles should be shown next to Figure 5.*

We do mention in section 4.1 that dust emissions increase the aerosol number concentration by more than 5,000 cm$^{-3}$ over remote deserts. Following the reviewer's suggestion, we have also added a figure in the revised manuscript (Figure 6c) to show the changes in aerosol number concentration after switching on/off the mineral dust emissions. Due to the addition of mineral dust, total aerosol number concentration increases over the deserts, especially over remote deserts such as Taklimakan and Atacama, and decreases downwind of them and over polluted areas due to the coagulation of the coarse dust particles with the smaller anthropogenic aerosols.

3. *There is no mention in the paper about the geographic distribution of soluble and insoluble fractions in the dust modes predicted by the model. This would definitely help with the discussion and interpretation of the results. A map showing this distribution would help understanding the underlying processes.*

Thank you for the good suggestion; we have added a figure showing the spatial distribution of the insoluble fraction of particles.

4. *The paper does not explain how the CDNC shown in the maps is calculated. Are those grid-cell averages? Are those in-cloud values? Is this the value only after activation subroutine is called? Or are these values produced by the full cloud-microphysical scheme?*

CDNC values reported in the manuscript are referred to the number concentration of droplets nucleated in-cloud (i.e., right after the activation subroutine is called) and represent an upper limit since droplet depletion by collision, coalescence and collection are not taken into account. This information has been added in section 3.1

5. *No indication of the frequency of occurrence of liquid clouds at the level in the model is mentioned, nor that of the climatological cloud cover in those regions. If this is somehow included in the manuscript, the overall importance of dust on CDNC globally could be better assessed. The specifics of the annual average CDNC shown in the paper should be discussed and described in detail.*

The annual average low level cloud cover calculated by the EMAC model has been added in figure 2. While the calculated cloud cover over the main deserts is low (i.e., typically lower than 5%), CDNC is also sensitive to mineral dust emissions far from its sources and over areas with high cloud cover (e.g., over Europe and Eastern Asia). This is now discussed in the revised text.

6. *Some fundamental issues with the unified theory should be discussed by the authors in this manuscript. In particular the potential oversimplification of the activation process for insoluble particles with small soluble coatings (as could potentially be the case for dust particles). See specific comments.*

As discussed in detail below, we always assume that insoluble material (i.e., mineral dust) is expressed by the FHH terms. The aerosol hygroscopicity ($\kappa$) of the soluble fraction is calculated according to the simple mixing rule. Then, based on the FHH terms, the $\kappa$ hygroscopicity and the insoluble fraction ($e_i$), the exponent $x$ in Eq. 4 is calculated with a power law fit between $S_g$ and $D_{dry}$ as described in Kumar et al. (2011a). $x$ lies between -0.86 for $e_i$=1 and -1.5 for $e_i$=0.

7. *I suggest modifying some of the conclusions of the paper, since they can be overreaching. It doesn't seem that the paper actually "demonstrates" that the biases are substantial, or that this treatment is indeed correct. In fact, the authors acknowledge almost no sensitivity of CDNC to massive cuts in dust emissions, or hydrophilicity parameter, or on dust chemical composition. For example, I quote "By assuming drastic differences in the dust source and the dust hydrophilicity we find only small (∼5%) changes in the average CDNC".*

The CDNC changes reported in the conclusion section are global averages. The global average changes of CDNC are small, mainly due to the negligible changes over the oceans and in some cases due to counteracting effects (i.e., opposite response of CDNC over the deserts and downwind of them). However, larger CDNC changes are calculated regionally (i.e., up to 30% over the deserts and 10% over highly polluted areas). This is now emphasized in the conclusions as well.

**Specific comments**
8. *Section 2.2. Line 265. It is not clear from the equations nor the references cited in the document, how can an exponent x = -3/2 be obtained from equation (3) when there is insoluble material but no FHH terms. The -3/2 exponent arises from the fact that the whole volume of the particle contributes to the soluble material during the activation process. It is not explicit from the document what is the expression relating critical diameter and critical supersaturation when there is a substantial fraction of insoluble material (i.e., in equation 3, with no FHH terms, but a small amount of soluble material). The relations between dry aerosol size and critical supersaturation are severely modified when an insoluble core is present (see for example, Pruppacher and Klett, chapter 6, equations 6-37 to 6-42). Therefore, there is a possibility that one could see substantial changes in CDNC by simply improving the description of the relation between critical diameter and critical supersaturation for cases where there is an insoluble core (no FHH terms). This issue should be explored and discussed in the paper.*

We always assume that insoluble material (i.e., mineral dust) is expressed by the FHH terms. When mineral dust is present in the soluble modes, $x$ is calculated by

performing a power law fit between $s_g$ and $D_{dry}$ as described in Kumar et al. (2011a) and is given by:

$$x=x_{FHH}*exp(log(-1.5/x_{fhh})*(1-e_i)0.1693*exp(-0.988\kappa)$$

$x$ lies between $x_{FHH}$ for $e_i=1$ and -1.5 for $e_i=0$. $e_i$ is the fraction of mineral dust in the mode, $\kappa$ is the total aerosol hygroscopicity of the soluble fraction of the mode and the $X_{FHH}$ depends on $A_{FHH}$ and $B_{FHH}$ used (Kumar et al., 2009b) and here is equal to -0.86.

Black carbon, which can exist in the soluble modes of our model after coagulation, is assumed to be part of the soluble material and affects the total aerosol hygroscopicity of the soluble fraction according to the simple mixing rule but not the exponent x of the soluble particle which, in the absence of mineral dust, is equal to -1.5.

9. *It would be convenient for the readers to see average values of CDNC, or average fractional changes printed in the global maps of figures 1, 2, 5, 6, 7 and 8*

    This information has been added to the figures.

10. *The difference between results shown in Figure 5, and Figure 8c are not entirely clear to me. So figure 5 has no mineral dust emissions, and Figure 8c, was performed with 50% aerosol emissions compared to base case? So in the case of no emissions, there is a net decrease in CDNC, but when there is only a 50% decrease in the emission load there is an increase in CDNC?*

    They are just illustrated vice versa. Figure 5 depicts the CDNC change after including mineral dust emissions (increasing mineral dust) while figure 8c depicts the CDNC change after assuming 50% less dust emissions (decreasing mineral dust). In both cases CDNC decreases with increasing mineral dust.

11. *Line 712. Should it read "insensitive"?*

    Changed.

12. *Figure 2. Are these values grid-cell averages? Or are they in-cloud values only?*

    They are in-cloud values. We have added this information in figures 1, 2 and 3.

13. *From figure 8, it seems that BFHH parameter has a larger (or at least comparable) impact to reducing mineral dust emissions by 50%? This should be discussed in much more detail. As mentioned above, perhaps showing the net impact that the 50% reduction*

*in dust emissions has on aerosol number concentration would be helpful in the interpretation of the results.*

In both simulations, the sensitivity of CDNC is dominated by the changes in the calculated critical supersaturation of the particle as well as the exponent $x$ in Eq. (4). Reducing the dust emissions by 50% results in an increase of aerosol number concentration by less than 10% downwind of deserts and over polluted regions (which mostly control the global average change of CDNC). However, the insoluble fraction of particles over these regions decreases by 40% which significantly affects, through changes in equilibrium water vapor supersaturation (Eq. 3), the "CCN spectrum" (Eq. 4). Similarly, increasing the hydrophilicity of the dust particles by changing the $B_{FHH}$ parameter, directly affects the equilibrium supersaturation and the "CCN spectrum" through changes in the exponent $x$. These issues are discussed in sections 5.2 and 5.3.

14. *Similarly, the paper shows very little sensitivity of CDNC to dust chemical composition, but relatively high sensitivity to the BFHH parameter. However, it is reasonable to believe that the FHH parameters are linked to the chemical composition of the mineral dust particles. Therefore, some discussion should be included regarding the relationship between the FHH theory parameters and dust chemical composition, and the potential impacts it could have in the simulations.*

The sensitivity test presented in section 5.1 describes the effect of the chemical composition of dust on the results only due to changes on the thermodynamic interactions with inorganic anions. The FHH parameters describe the hydrophilicity of fresh dust. Their values are determined to reproduce the measured CCN activity of the dust samples. Kumar et al. (2011b) tested the CCN activity of aerosols dry generated from clays, calcite, quartz, and desert soil samples from Northern Africa, East Asia/China, and Northern America. They found that $B_{FHH}$, which strongly affects the equilibrium curve, varied from 1.12 to 1.30 (i.e., ±10% from 1.2 which is the value used in our base case simulation). Therefore, the sensitivity test presented in section 5.2, where we assumed 10% lower $B_{FHH}$, can represent the potential impacts on the results due the simplification of using a globally uniform set of FHH parameters to describe the hydrophilicity of mineral dust independently of its source and composition. Our results indicate that changes in the hydrophilicity of the freshly emitted dust, due to the variability of its composition with source region, can have an important impact on the calculated CDNC. This is now emphasized in section 5.2.